# Experimental mining plumes and ocean warming trigger stress in a deep pelagic jellyfish

Vanessa I. Stenvers [1,2,6] ✉, Helena Hauss [1,3,6], Till Bayer [1],
Charlotte Havermans [4], Ute Hentschel [1], Lara Schmittmann [1],
Andrew K. Sweetman[5] & Henk-Jan T. Hoving [1]

The deep pelagic ocean is increasingly subjected to human-induced environmental change. While pelagic animals provide important ecosystem functions including climate regulation, species-specific responses to stressors remain poorly documented. Here, we investigate the effects of simulated ocean warming and sediment plumes on the cosmopolitan deep-sea jellyfish *Periphylla periphylla*, combining insights gained from physiology, gene expression and changes in associated microbiota. Metabolic demand was elevated following a 4 °C rise in temperature, promoting genes related to innate immunity but suppressing aerobic respiration. Suspended sediment plumes provoked the most acute and energetically costly response through the production of excess mucus (at ≥17 mg L$^{-1}$), while inducing genes related to aerobic respiration and wound repair (at ≥167 mg L$^{-1}$). Microbial symbionts appeared to be unaffected by both stressors, with mucus production maintaining microbial community composition. If these responses are representative for other gelatinous fauna, an abundant component of pelagic ecosystems, the effects of planned exploitation of seafloor resources may impair deep pelagic biodiversity and ecosystem functioning.

Since the deep pelagic ocean lies far beyond our natural reach, the impacts of human-induced environmental change were long thought to be of minor significance[1]. The pelagic ocean, stretching below the surface to the seabed, comprises the largest habitat on earth and accounts for over 90% percent of the ocean's livable volume[2–4]. While we have explored less than 1% of this vast realm[4,5], it is becoming ever more apparent that human activities have pronounced effects on pelagic ecosystems. Increasing evidence shows that stressors such as ocean warming, resource exploitation, deoxygenation, acidification, and pollution affect the entire water column[4,6–9]. Moreover, the commercial mining of deep-sea mineral resources is becoming a rapidly approaching reality despite many unanswered questions about the environmental impacts[10,11]. Up to now, most research on the effects of human-induced stressors on deep-sea ecosystems has focused on seafloor environments, paying less attention to organisms in the water column[4,5,11,12].

A major threat to pelagic fauna is the global increase in ocean temperatures that is caused by the emission of greenhouse gases. Deep pelagic organisms have evolved under relatively stable thermal conditions, which generally translates to narrow physiological tolerances that make them particularly vulnerable to environmental change[4]. While organisms adapted to undergo vertical migrations may

¹GEOMAR, Helmholtz Centre for Ocean Research Kiel, Wischhofstraße 1-3, 24148 Kiel, Germany. ²Department of Invertebrate Zoology, National Museum of Natural History, Smithsonian Institution, Washington, DC P.O. Box 37012, USA. ³Norwegian Research Centre AS (NORCE), Stavanger, Norway. ⁴HYIG ARJEL, Functional Ecology, Alfred Wegner Institute Helmholtz Centre for Polar and Marine Research, Am Handelshafen 12, 27570 Bremerhaven, Germany. ⁵Seafloor Ecology and Biogeochemistry Research Group, Scottish Association for Marine Science (SAMS), Oban, Scotland, UK. ⁶These authors contributed equally: Vanessa I. Stenvers, Helena Hauss. ✉e-mail: vstenvers@geomar.de

be less sensitive to changes in temperature than non-migrant species[13], metabolic activity in ectotherms generally scales with temperature. As a result, increasing temperatures can directly increase metabolic rates and thus the nutritional demand of pelagic organisms[14]. Since food in the deep sea is generally scarce, elevated temperatures could therefore increase metabolic demands beyond normal energy intakes. This offset of metabolic balances may be further amplified by the depletion of dissolved oxygen, which is an accompanying effect of warming oceans[15]. Although the scaling of metabolic rates in ectotherms by temperature is well characterized for a wide range of pelagic fauna[13,16], data remains limited for fragile or rare taxa as many deep pelagic animals are difficult to study and collect from their natural environment[3].

In addition to ocean warming, the mining of deep-sea mineral resources has recently raised concerns about the potential negative effects on pelagic animals[11]. Although the commercial exploitation of the seabed has yet to commence, pilot studies for the extraction of ore minerals are already underway[10]. Few regulations exist to guide future mining operations in minimizing the effects on pelagic ecosystems, despite control of the International Seabed Authority (ISA) that regulates all mineral-related activities in international waters[11,12,17,18]. One of the risks to pelagic ecosystems will be the resuspension of sediments by collector vehicles on the seabed and the discharge of sediments into the water column after separation from the ore minerals[19]. Depending on the release depth and duration of the mining operation, the generated sediment plumes can affect the entire water column, disperse for hundred kilometers, and last several years[20,21]. Particle size and physical ocean processes may further affect the duration of sediment suspension[22]. Although there are few studies on the response of seafloor communities to sediment plumes, species and community responses of pelagic animals have not been studied[12,19]. Nevertheless, hypothesized effects on pelagic biota include the obstruction of respiratory organs, olfactory senses, and feeding structures. Moreover, sediment particles may adhere to animals and reduce buoyancy, clog surfaces or be ingested[10,11,17]. This not only affects the animals themselves but could potentially disturb the microbial communities living in association with them. While only few studies have focused on the microbiomes of pelagic animals, most demonstrate host-specific communities that are thought to play important roles in host immune systems, nutrition or development[23,24]. Compromised microbial communities can therefore be detrimental to host survival[25]. The effects of sediment plumes on pelagic animal behavior, physiology, and associated microbiomes need investigation.

Both warming and deep-sea mining are expected to lead to a significant loss in biodiversity of pelagic animals and associated ecosystem services[4]. This is of particular concern as deep pelagic fauna play a vital role in the provisioning of commercially important fish stocks such as tuna, nutrient recycling, and atmospheric carbon sequestration[4,26–28]. Pelagic animals are mainly fueled by primary productivity from the upper water layers, either directly by migrating to the surface at night to feed on phytoplankton and grazers, or indirectly from sinking particles and predation on lower trophic levels[4,26]. In doing so, they redirect carbon to greater depths via the release of organic matter (e.g. fecal material, mucus), via respiration at depth or as sinking carcasses, collectively known as the biological carbon pump (BCP)[14,27,29]. By preventing carbon from re-entering the atmosphere, pelagic organisms contribute to the regulation of our climate[30]. As a consequence, pelagic ecosystems are intricately linked to terrestrial habitats, and alterations of their services will affect environments beyond the sea.

To implement effective mitigation efforts, research to determine threshold sensitivities of pelagic animals to environmental change is critical yet challenging. Pelagic organisms are not only difficult to collect from their environment but also to maintain in the laboratory as many have specific light, depth or dissolved oxygen preferences[31].

A large part of pelagic communities comprises gelatinous zooplankton, spanning a variety of taxa that are characterized by a high water content (>95%) and fragile tissues. Well-known examples include cnidarian medusae, many of which inhabit coastal or surface waters. Less familiar are the medusae living in the deep sea. Since their distribution is often patchy and their fragile bodies are easily damaged by net collections[3], responses to stressors in jellyfish have so far only been studied in coastal species (e.g.[32,33]). Given that sediment suspension in the deep ocean is generally low and temperatures at depth are buffered, we hypothesize that deep-sea animals, including medusae, are highly sensitive to disturbance.

Here, we investigate the effects of simulated ocean warming and mining-induced sediment plumes on the pelagic helmet jellyfish *Periphylla periphylla* (Péron & Lesueur, 1810). This cosmopolitan medusa occurs from the surface down to 4000 m depth[34,35], making it a representative organism to study the effects of warming and mining in the global deep ocean. Moreover, *P. periphylla* is known for its high abundance in several Norwegian fjords, allowing for relatively easy and gentle collection of individuals while their biology and physiology remain relevant for oceanic populations[36,37]. By exposing medusae from these fjords to a range of increasing temperatures and abyssal sediment concentrations during ship-board experiments, we collect the first physiological and molecular data of a deep-water jellyfish in response to anthropogenic stressors. Specifically, we measure their stress response based on physical condition, respiration, ammonium excretion, gene expression and changes in associated microbiota. Our results show that *P. periphylla* can invoke several physiological compensation strategies involving metabolic pathways related to innate immunity, respiration, and wound repair. While our temperatures treatments are likely most energetically costly in the long run, short-term exposure to suspended sediment caused acute physical effects through the production of excess mucus. Microbial community composition appeared unaffected by both stressors, with mucus production being an effective yet costly strategy to maintain bacterial composition during simulated sediment plumes.

## Results

In total, 64 *P. periphylla* were collected from the Lurefjord (*n* = 38) and Sognefjord (*n* = 26), of which 21 were experimentally exposed to a range of increasing temperatures and 43 to different concentrations of suspended sediment (Supplementary Table 1). For the gene expression analysis, transcriptomes of 29 medusae were successfully sequenced (i.e. $n_{temp}$ = 11 and $n_{plume}$ = 18), producing an average 35.7 (±1.1 SD) million read pairs per sample following initial quality filtering steps. The final de novo transcriptome for *P. periphylla* comprised 839,298 transcripts and a total of 97.8% metazoan orthologs were recovered from the assembled transcriptome (Supplementary Table 2). Within the transcriptome (also containing gene isoforms and non-coding RNA), up to 21,429 transcripts were actively expressed, which we hereafter refer to as putative genes. For analysis of microbiomes, 33 medusae were successfully sequenced (i.e. $n_{temp}$ = 12 and $n_{plume}$ = 21), in addition to 12 seawater and 3 sediment reference samples.

### Response to warming temperatures

Metabolic activity in *P. periphylla* appeared to be mainly upregulated by the four-degrees rise in temperature. Although no significant differences could be detected in respiration rates among treatments (ANOVA df=2, F = 1.354, *p* = 0.29, *n* = 17), average values nearly doubled when comparing individuals at 7.5 and 11.5 °C (i.e. 0.013 ± 0.007 SD versus 0.025 ± 0.010 mg $O_2$ g $WW^{-1}$ $h^{-1}$; Fig. 1a). Ammonium excretion was significantly elevated in *P. periphylla* exposed to 11.5 °C (ANOVA df=2, F = 7.65, *p* = 0.004, *n* = 29), increasing over six-fold when compared to individuals at 7.5 and 9.5 °C (Tukey p-adj.<0.007; i.e. 0.185 ± 0.155 versus 0.030 ± 0.055 and 0.028 ± 0.027 μmol $NH_4$ g $WW^{-1}$ $h^{-1}$ respectively; Fig. 1b).

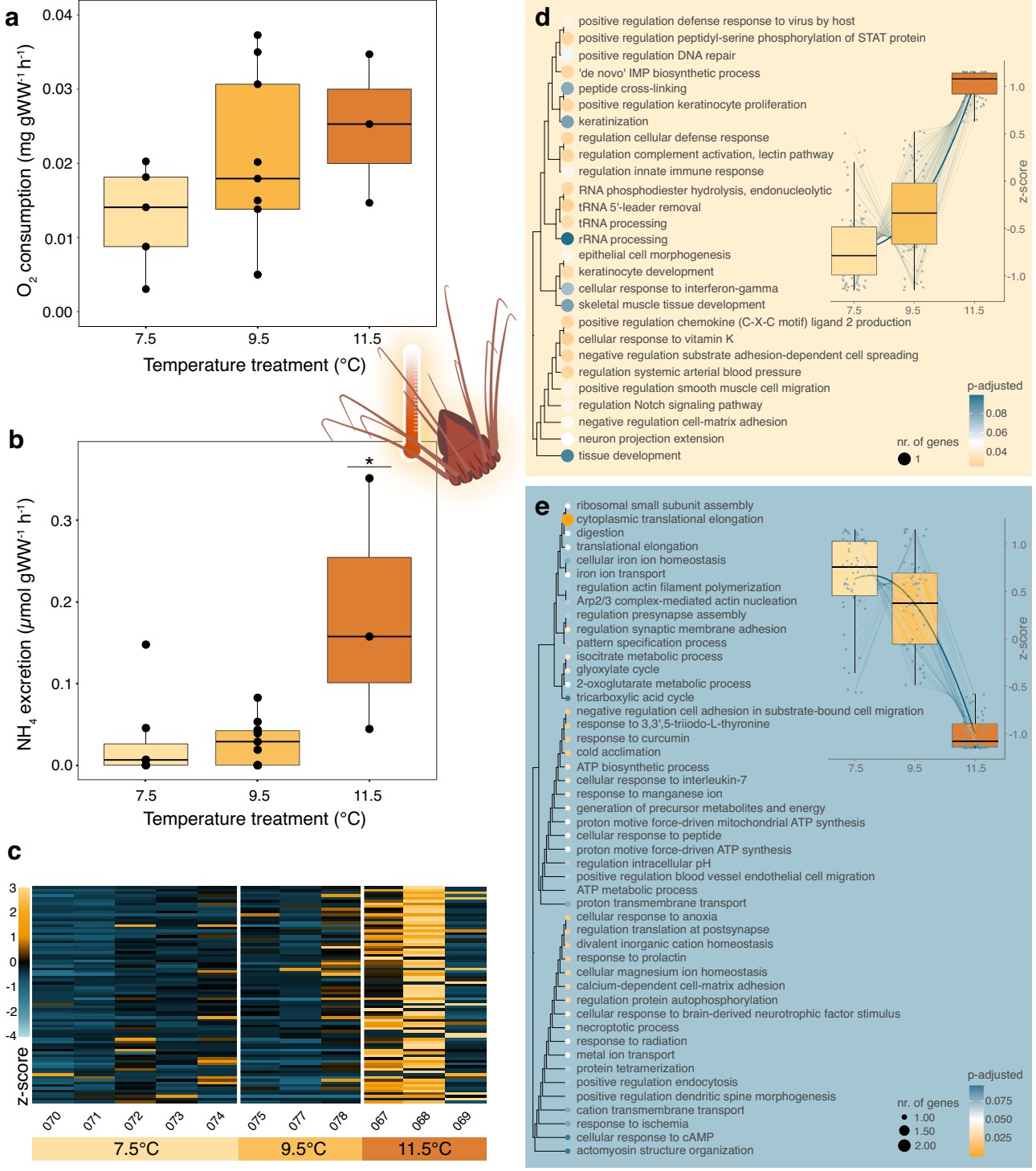

Fig. 1 | **Metabolic response of *Periphylla periphylla* to a range of increasing temperatures.** **a** Respiration rates measured directly from oxygen optodes (*n* = 17 biologically independent samples). **b** Ammonium (NH₄) excretion. Significant difference indicated by asterisk (*p* = 0.004, *n* = 21 biologically independent samples) analyzed with a one-way ANOVA and post-hoc Tukey test. **c** Heatmap of 80 significantly differentially overexpressed genes, comparing all temperature treatments using the likelihood ratio test (LRT). Columns indicate sample numbers. Each row represents one gene, with colors showing z-score-transformed expression values where blue indicates below- and yellow above-average expression. **d**, **e**

Gene ontology (GO) for biological processes of significantly expressed genes in *P. periphylla* (identified with the LRT; *n* = 11 biologically independent samples), showing enriched GO terms for **d** overexpressed and **e** underexpressed genes. Dot size in the GO tree indicates the number of genes, while colors indicate p-adjusted values (yellow <0.05, white 0.05, blue > 0.05). Box plots indicate changes in expression across treatments, taken from Supplementary Fig. 1a. All box plots show median (central line) and interquartile ranges (IQR) with whiskers extending to 1.5x the IQR range. Source data are provided as a Source Data file.

The strongest metabolic response of *P. periphylla* at 11.5 °C was substantiated by our differential gene expression results. In total, 20,697 putative genes were expressed in the analyzed tissue, of which 162 were significantly differentially expressed (DE). Of these, nearly half ($n = 80$) showed marked overexpression in the 11.5 °C treatment (Fig. 1c, Supplementary Fig. 1a, group 1). By assigning biological function through the Kyoto Encyclopedia of Genes and Genomes (KEGG)[38], functional categories of overexpressed genes were found to be associated with processes such as ribosome translation, the immune system (i.e. C-type lectin mediated by Q9NBX4, a reverse transcriptase), and cell growth and death (Supplementary Table 3, T1). Our gene ontology (GO) analysis further confirmed that the significantly upregulated genes were enriched in processes related to innate immunity (i.e. activation of the complement system by Q7SIC1, a defensive fucolectin; and O75369, a filamen-B), and the regulation of the cellular response to viruses or DNA repair (i.e. both mediated by P0DP91, an excision repair protein, and Q8N328 a transposable element; Fig. 1d). Other enriched genes were related to the processing of genetic information (e.g. ribosomal rRNA and transfer tRNA), cell migration and adhesion (i.e. involving chemokines and cell spreading mediated by Q62009, an extracellular matrix protein).

Similar to overexpressed genes, downregulated genes showed a highest metabolic change in *P. periphylla* exposed to 11.5 °C (Supplementary Fig. 1a, group 2). In total, 54 genes were significantly underexpressed with KEGG pathway maps revealing an involvement in processes such as translation, transcription, cell growth and death, energy metabolism involving oxygen (i.e. citrate cycle mediated by Q9Z2K9, an isocitrate hydrogenase, and oxidative phosphorylation by Q03105, P10719 and P06576, a proton ATPase and two ATP synthases), and the immune system (i.e. Fc gamma R-mediated phagocytosis by Q8AVT9, an actin-binding protein; Supplementary Table 3, T2). Our GO analysis further showed enrichment related to the cellular response to anoxia (e.g. ischemia, resulting from a decline in oxygen levels, mediated by P70531, a factor-2 kinase), cold acclimation and energy metabolism on a cellular level (i.e. both involving adenosine triphosphate, ATP, or citric cycle by P10719 and P06576, two ATP synthases; Fig. 1e). Moreover, it should be mentioned that two significantly underexpressed genes matched proteins of cnidarian origin, including a ribosomal protein (P38984) and MAM/LDL-receptor (B3EWZ6). The contrasting response of *P. periphylla* in the 11.5 °C treatment was substantiated by our principal component analysis (PCA), where separation of treatments was mostly visible along the second principal component axis (14.9%) and where transcriptomes at 11.5 °C clustered most distinctively from other treatments (Supplementary Fig. 2a).

When comparing the microbial community structure of *P. periphylla* across treatments, jellyfish microbiomes were significantly different from the sea water (PERMANOVA df=1, F = 23.20, $p = 0.001$, $n = 24$) with no effect of the temperature treatments (Supplementary Fig. 4a, Supplementary Table 4). Nevertheless, the sampling date proved to significantly affect microbial composition (PERMANOVA df=1, F = 8.28, $p = 0.004$, $n = 12$) since not all *P. periphylla* were collected on the same date due to logistic limitations aboard the research vessel. Consequently, microbiomes of individuals exposed to 9.5 °C could not be compared to those at 7.5 and 11.5 °C, as individuals in the former treatment were collected on different day. Moreover, no statistical differences were found between microbiomes at 7.5 and 11.5 °C (PERMANOVA df=1, F = 1.15, $p = 0.306$, $n = 8$; Supplementary Fig. 4b). When comparing the average number of observed amplicon sequence variants (ASVs) across treatments, most ASVs were recovered from individuals at 7.5 °C (average $210 \pm 67$, $n = 5$), followed by the 9.5 °C ($145 \pm 65$, $n = 4$) and 11.5 °C treatments ($158 \pm 3$, $n = 3$; Supplementary Fig. 3a). Similarly, average diversity of ASVs estimated through the Shannon index proved to be lowest at 11.5 °C (Supplementary Fig. 3b).

For the *P. periphylla* exposed to 7.5 and 11.5 °C, the most dominant ASVs belonged to the phyla Proteobacteria (83.4% and 77.7%, respectively), Campilobacterota (11.4% and 6.7%) and Firmicutes (1.2% and 6.8%; see below). Individuals exposed to 9.5 °C were dominated by ASVs within the Firmicutes (37.3%), Bacteroidota (26.5%) and Proteobacteria (21.2%). Similarly, seawater samples were dominated by Proteobacteria (38.4%) and Bacteroidota (8.2%), but also showed elevated numbers of Chloroflexi (10.1%) which only occurred in relatively low numbers on *P. periphylla* (i.e. up to 1.2%). A total number of 2390 unique ASVs were recovered from seawater samples ($n = 12$, average $577 \pm 144$ SD), of which 152 (6.4%) were shared with *P. periphylla*.

## Response to simulated sediment plumes

Simulated sediment plumes showed a marked effect on *P. periphylla*'s physical condition. While the doubling of respiratory rates for individuals in the 0 and 333 mg L$^{-1}$ treatments was not statistically different (i.e. $0.012 \pm 0.006$ SD versus $0.028 \pm 0.013$ mg O$_2$ g WW$^{-1}$ h$^{-1}$; ANOVA df=4, F = 0.911, $p = 0.473$, $n = 29$; Fig. 2a), suspended particle load had a visible negative effect on *P. periphylla* at concentrations >17 mg·L$^{-1}$. After 1.5 hours, particles started aggregating on the bell and lappets (i.e. edges of the bell) and the jellyfish started to produce excess mucus that slowly sloughed off (Supplementary Fig. 5). Effects persisted throughout the entire incubation period, with *P. periphylla* exposed to 33, 167 and 333 mg L$^{-1}$ of suspended sediment showing significantly lower health scores compared to those in the 0 and 17 mg L$^{-1}$ treatments (ANOVA df= 4, F = 42, $p = 0.003 \cdot 10^{-4}$, $n = 43$ biologically independent samples examined over $n = 18$ treatment groups; Tukey p-adj.<0.008; Fig. 2b). Moreover, all health ratings beyond 6 hours of incubation, with the exception of the control, were significantly lower than health ratings recorded at the start of the experiments (ANOVA df=4, F = 35.1, $p = 0.002 \cdot 10^{-13}$, $n = 90$ timepoints examined from $n = 18$ treatment groups; Tukey p-adj.=0.000).

The observed increase in mucus production appeared to be matched by the overexpression of genes within the two highest sediment treatments (Fig. 2c). In total, 21,429 putative genes were expressed, of which 291 genes were found to be significantly DE. Nearly half of these were specifically overexpressed in the 167 and/or 333 mg L$^{-1}$ sediment treatments ($n = 121$; Supplementary Fig. 1b group 3 and 2). KEGG pathway maps (Supplementary Table 3) showed involvement of these genes in processes such as energy metabolism requiring oxygen (i.e. oxidative phosphorylation and thermogenesis by P10719 and P06576, two ATP synthases), translation, xenobiotic biodegradation (i.e. by P19804, a nucleoside diphosphate kinase), catabolism, cell growth and death, focal adhesion, and the immune system (complement and coagulation cascades by D3YXG0, a hemicentin, and B3EWZ3, a coadhesin of cnidarian origin). Our gene ontology (GO) analysis further confirmed enrichment in processes related to proteolysis (e.g. Q9CQ52 and Q3SYP2, two proteases, and P55115, an extracellular zinc metalloproteinase), ATP energy synthesis (i.e. by P10719 and P06576) and muscle movement (i.e. GO categories related to actin and the regulation of membrane repolarization during muscle cell action potential; Fig. 2d). Other enriched genes were associated with muscle cell development and processing of genetic information (e.g. DNA recombination and biosynthetic process). Interestingly, few genes beyond the 50 enriched genes shown in Fig. 2d were associated with microfilament assembly (i.e. actin) and wound healing (i.e. by Q8BTM8, a filament-A involved in cell migration and adhesion, and D3ZHA0, a filament-C involved in cytoskeletal organization; Supplementary Fig. 6). Genes related to genetic information processing and epithelial morphogenesis (i.e. by Q9ERK0, a serine/threonine-protein kinase) were specifically overexpressed in the highest sediment concentration (Supplementary Fig. 1b, Group 2). Moreover, three significantly overexpressed genes could be matched to cnidarian proteins, including a ribosomal protein (P38984), a

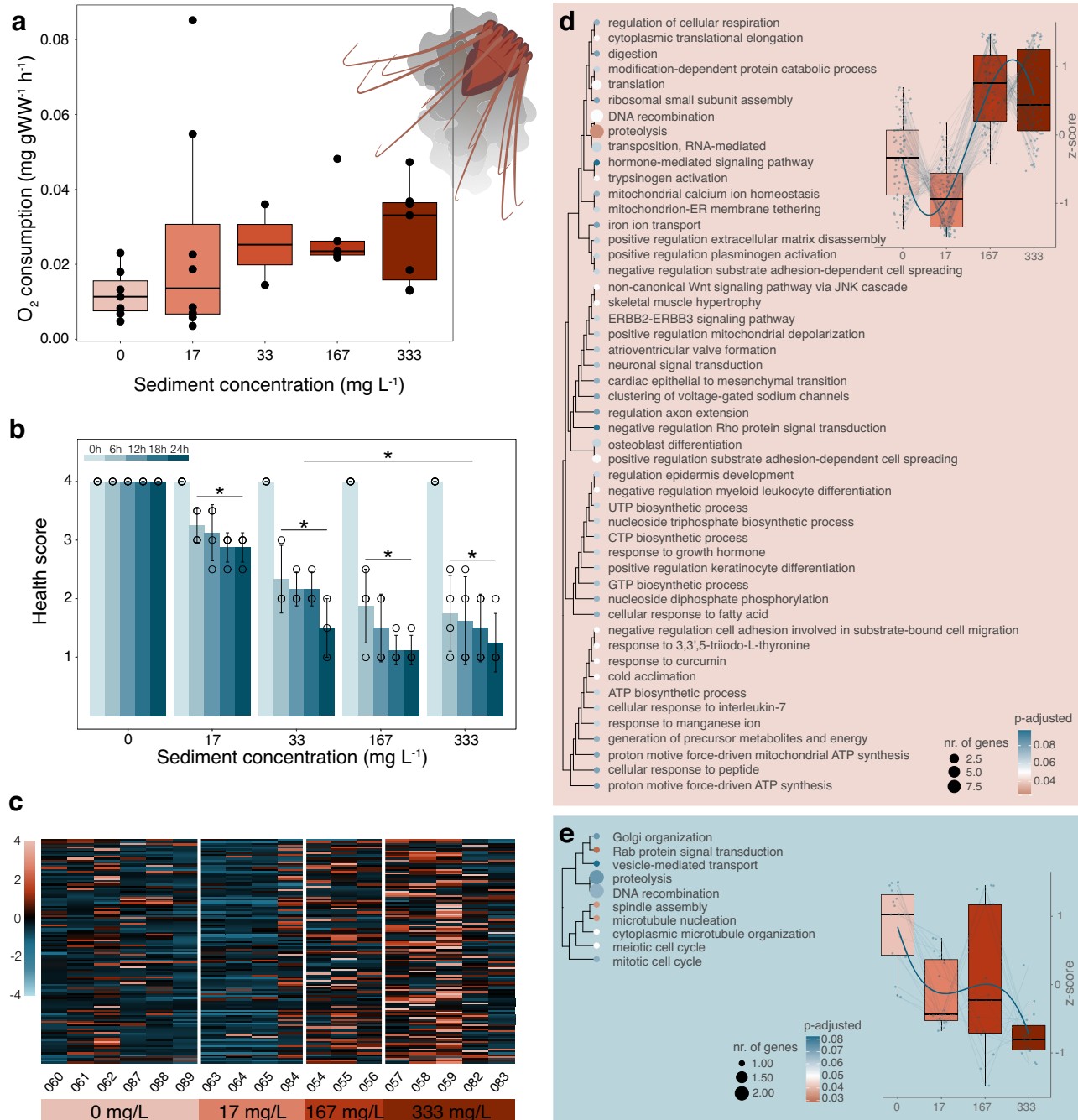

**Fig. 2 | Metabolic response of *Periphylla periphylla* to simulated sediment plumes. a** Respiration in response to suspended sediment as estimated from Electron Transfer System (ETS) activity ($n = 29$ biologically independent samples). **b** Average health rating of *P. periphylla*, from 0 h (light blue) to 24 h (dark blue). Significant differences indicated by asterisk, analyzed with one-way ANOVA and post-hoc Tukey tests between treatments ($p = 0.003 \cdot 10^{-4}$, $n = 43$ biologically independent samples examined over $n = 18$ treatment groups) and among timepoints within treatments ($p = 0.002 \cdot 10^{-13}$, $n = 90$ timepoints from $n = 18$ treatment groups). Data are represented as mean +/- SD. **c** Heatmap of 121 significantly differentially overexpressed genes, comparing all treatments simultaneously using the likelihood ratio test (LRT). Columns indicate sample numbers. Each row represents one gene, with colors showing z-score-transformed expression values where blue indicates below- and yellow above-average expression. **d, e** Gene ontology (GO) enrichment for biological processes of significantly expressed genes in *P. periphylla* (identified with the LRT; $n = 18$ biologically independent samples), showing enriched GO terms for **d** overexpressed genes (tree trimmed to show the 50 first terms) and **e** underexpressed genes. Dot size in the GO tree indicates the number of genes, while colors indicate p-adjusted values (red <0.05, white 0.05, blue > 0.05). Box plots indicate changes in expression across treatments, taken from Supplementary Fig. 1. All box plots show median (central line) and interquartile ranges (IQR) with whiskers extending to 1.5x the IQR range. Source data are provided as a Source Data file.

protease inhibitor (A0A6P8HC43), and a toxin (i.e. P0DN18, known as U-actitoxin-Avd3q).

In contrast, 15 genes were found to be gradually downregulated with increasing sediment concentrations (Supplementary Fig. 1b group 6). These genes were associated with the KEGG pathways involving metabolism (e.g. nucleotide metabolism), sensory systems, calcium signaling and renin secretion. Additionally, our GO analysis showed enrichment in processes related to cellular organelles and structural components of the cell (Fig. 2e), while one significantly underexpressed gene could be matched to a cnidarian MAM/LDL-

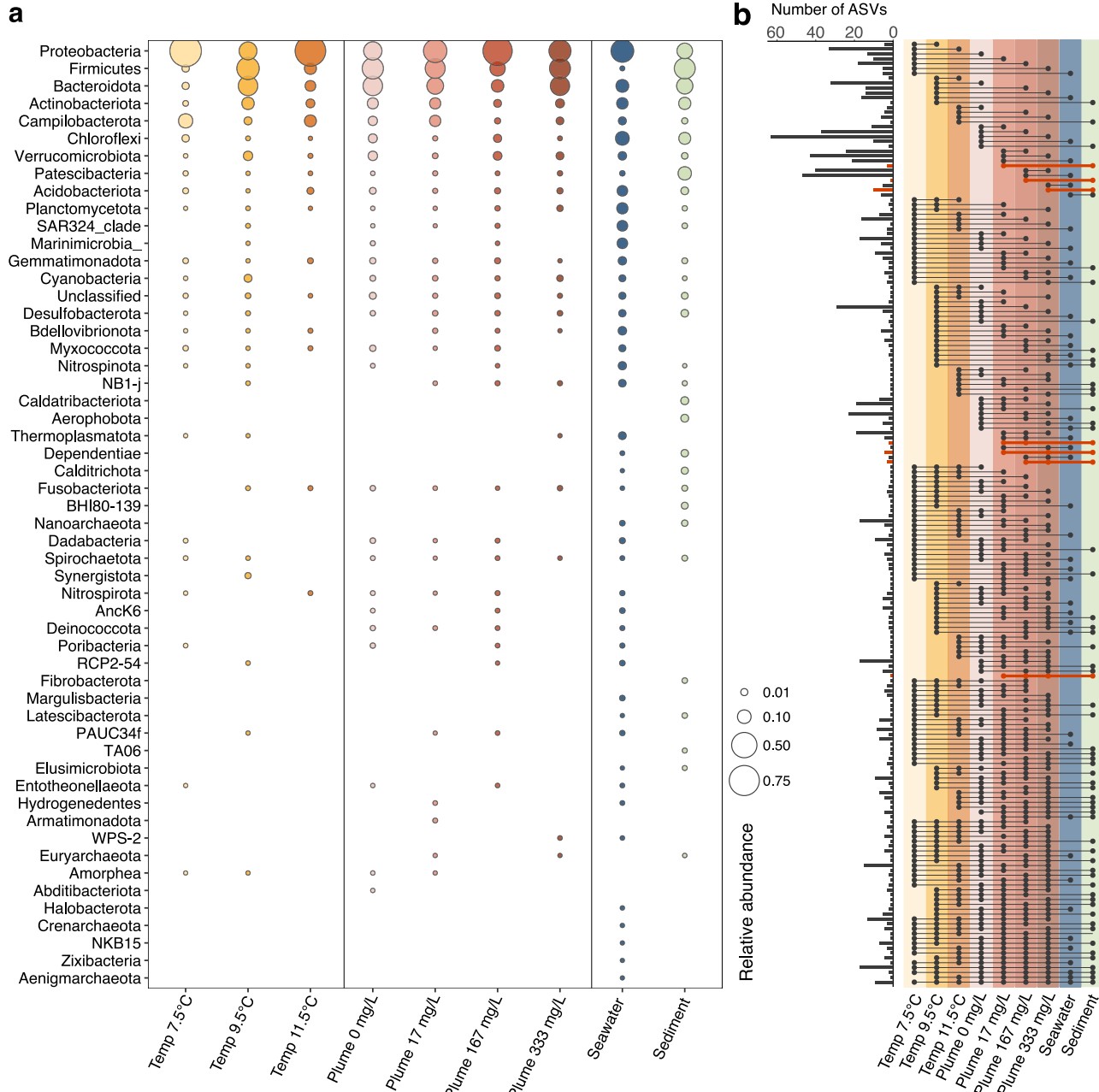

**Fig. 3 | Microbial community composition of *Periphylla periphylla* associated with the outer bell, seawater and sediment. a** Relative microbial community composition showing all phyla across the simulated temperature ('temp', left) and sediment ('plume', middle) treatments, compared to seawater and sediment reference samples (right). **b** Shared amplicon sequence variants (ASVs) across experiments, treatments, and controls. Presence across treatments is indicated by dots and lines (right panel), with the number of ASVs from each combination visible in the left panel. ASVs exclusively shared between sediment and simulated plume treatments (17, 167, and 333 mg L$^{-1}$) are highlighted in red. Source data are provided as a Source Data file.

receptor (B3EWZ6). From our PCA, separation of treatments was mostly visible along the second principal component (8.7%) with transcriptomes of *P. periphylla* exposed to 17 and 333 mg L$^{-1}$ clustering most distinctively (Supplementary Fig. 2).

Microbial community structure of *P. periphylla* in response to suspended sediment appeared to be differently affected among our two sampling days and thereby experimental runs (PERMANOVA df=3, F = 3.9, p = 0.001, n = 21; Supplementary Table 5). While *P. periphylla* in the first round of experiments showed significantly different microbiomes at 333 mg L$^{-1}$ of suspended sediment compared to the other treatments (PERMANOVA df=3, F = 4.2, p = 0.018, n = 12), those in the second experimental round shared similar community profiles across treatments (Supplementary Fig. 4c). Highest average ASVs numbers

were recovered in the 333 mg L$^{-1}$ treatment (444 ± 351, n = 4), followed by the 167 mg L$^{-1}$ (344 ± 87, n = 4) and 17 mg L$^{-1}$ treatments (153 ± 77, n = 7; Supplementary Fig. 3c), which was largely mirrored by the Shannon diversity index (Supplementary Fig. 3d). Similar to *P. periphylla* in the temperature experiments, microbiomes were significantly different from the seawater and sediment controls (PERMANOVA df=2, F = 20.11, p = 0.001, n = 21), with most abundant AVS belonging to the phyla Firmicutes (ranging from 13.8 to 32.7%), Bacteroidota (7.7 to 26.4%) and Proteobacteria (22.4 to 70.0%; Fig. 3a).

In addition to the observed microbial variability across dates, our results indicate the potential transfer of bacteria from the sediment to the *P. periphylla* outer bell microbiome. In total, 24 ASVs were exclusively shared between the sediment and jellyfish exposed to 17, 167 and

333 mg L[-1] of suspended sediment, excluding the control and seawater (Fig. 3b), belonging to the phyla Actinobacteriota, Bacteroidota, Euryarchaeota, Firmicutes, Fusobacteriota and Proteobacteria (Supplementary Table 6). Of these ASVs, 17 were recovered in *P. periphylla* exposed to 333 mg L[-1] of suspended sediment (i.e. in three out of four individuals in this treatment group), 7 ASVs in the 167 mg L[-1] treatments (i.e. in three out of four individuals) and 10 ASVs in the 17 mg L[-1] treatment (i.e. in four out of seven individuals). Moreover, the number of ASVs exclusively shared between the sediment and *P. periphylla* was highest in the first experimental round (i.e. 16 ASVs) with only one ASV shared in the second round.

## Discussion

In this study, we investigated the response of a deep pelagic jellyfish to simulated ocean warming and mining-induced sediment plumes, combining insights gained from physiology, gene expression and analysis of microbial symbionts. For the first time, we show how their biology may shift under stress, with implications for ecosystem functioning if this response is indicative of other pelagic fauna that play vital roles in atmospheric carbon sequestration, the cycling of nutrients, and the provisioning of food for commercially important fish stocks.

Our results show distinctive upregulation of metabolism in *P. periphylla* in response to a four-degree rise in temperature. Effects were most noticeable in the excretion of ammonium and gene expression, while the increasing trend in oxygen consumption could not be supported statistically. This moderate response in oxygen consumption can potentially be explained by *P. periphylla*'s ability to perform both aerobic and anaerobic respiration simultaneously[39], with downregulated genes here suggesting a shift away from aerobic metabolism. Although we did not find any significantly DE genes directly related to anaerobic metabolism (e.g. involving glycolysis), anaerobic respiration may have been continually active in *P. periphylla* across treatments. Consistent levels of anaerobic enzymes were also found in *P. periphylla* populations living at low versus higher oxygen concentrations, while those in low-oxygen environments carried significantly lower levels of aerobic enzymes[39]. Moreover, *P. periphylla*'s ability to respire anaerobically may contribute to the overall low respiration rates, which were within the range of previously reported values for *P. periphylla* and similar to those of other deep pelagic medusae capable of anaerobic respiration (Supplementary Table 7)[35,40]. Nevertheless, further research is required to determine whether oxygen concentrations and seawater temperature have similar effects on aerobic versus anaerobic metabolism in *P. periphylla*. Alternatively, the response in oxygen consumption may still become significant when sampling sizes are increased, considering the natural variability in these animals and associated data[35,40]. However, based on our current results and given that *P. periphylla* can vertically migrate to significantly warmer waters at night[34,35], we expect a relatively wide temperature tolerance in this species. Moreover, when considering current climate warming projections (i.e. approximately 1 °C over the next 84 years, with 4 °C warming only under extreme climate predictions)[41], global warming scenarios for the deep ocean do not appear to pose an immediate respiratory threat to *P. periphylla*.

The significant increase in ammonium excretion in *P. periphylla* in the highest temperature treatment likely resulted from an overall increase in temperature-driven metabolism, as ammonium is a direct byproduct of metabolic reactions[28]. Overexpressed genes related to DNA repair and ribosomal RNA (i.e. involved in the production of proteins) may specifically be linked to this response, as both were suggested to be general regulators of metabolism during environmental stress in different species of corals[42–44]. Although average excretion rates in *P. periphylla* exposed to 7.5 and 9.5 °C were similar to those reported for medusae from coastal and surface seas, rates of *P. periphylla* exposed to 11.5 °C were nearly twice as high (see

Supplementary Table 8 for summary)[28,45,46]. These results may indicate that *P. periphylla* is more sensitive to elevated temperatures in terms of ammonium excretion compared to other jellyfish.

Other metabolic costs of *P. periphylla* in response to the highest temperature treatment appeared to be associated with upregulation of innate immunity. Innate immune systems are found in most invertebrates and form a first line of defense to detect and target foreign molecules or damaged tissues[47]. In contrast, adaptive immunity has only been found in vertebrates, where the immune system may 'learn' from infections through the generation of antibodies targeting pathogens[48]. Since innate immunity is evolutionarily conserved across the animal kingdom, it has been used as a biomarker for stress or stress-tolerance in cnidarians such as corals and anemones[42,43,48–51]. Moreover, the immune response in *P. periphylla* may be further supported by the overexpressed genes related to chemokines, cell adhesion and cell migration. Although cell migration and adhesion perform important functions in a variety of processes, both are an integral part of immune responses that promote the movement of immune-related compounds and cells towards or away from affected areas[47,52]. Similarly, chemokines are known to promote immune responses through the recruitment of white blood cells[53]. The exact immune pathways and effector responses, however, are likely to be stressor- and species-specific. Little research exists on stressor-induced immune pathways in jellyfish, with gene expression of innate immunity so far only reported in the medusa *Aurelia coerulea* in response to physical disturbance[33]. While warming temperatures may compromise *P. periphylla*'s tissue integrity or increase susceptibility to pathogens, it is also possible that overexpression of immune-related compounds is a form of 'frontloading' to prepare for regularly encountered stress[51]. Since *P. periphylla* occurs at a variety of depths and thereby temperatures, the observed plasticity of expressed genes may indicate local adaptation to a wide depth distribution, that would also fit the observed respiratory response. In any event, our results show that short-term exposure to increasing temperatures has the marked potential to influence immune pathways in a deep pelagic jellyfish.

No significant changes were observed in the microbial communities associated with the outer bell of *P. periphylla* in response to our temperature treatments. This is in line with the thermal stability reported in several coral-associated microbiomes[54,55]. While our short-term temperature incubations had no effect on microbial community composition in *P. periphylla*, composition did differ with collection date. Since we only investigated microbiomes associated with the outer bell, it is possible that community composition was influenced by those from the surrounding water column and comprised a high proportion of transient, environmental bacteria. Although stressor-induced changes in microbiomes have not been investigated in jellyfish to the best of our knowledge, comparison of different body parts in coastal jellyfish showed highest variability of those associated with the outer bell[56,57].

Similar to *P. periphylla* under temperature treatments, oxygen consumption was only moderately affected by exposure to simulated sediment plumes. Yet where gene expression suggested a shift away from aerobic respiration under increasing temperatures, the two highest sediment treatments instead promoted genes associated with aerobic metabolism. Subsequently, exposure to suspended sediment clearly required a different metabolic response, likely caused by the production of excess mucus. Mucus production is a common stress response in cnidarians, including jellyfish, corals and anemones, where it acts as a physical and chemical barrier against pathogens or mechanical stress[58,59]. Moreover, excreted mucus is generally known to be energetically expensive, as it contains nitrogen-rich glycoproteins, lipids, nucleic acids and various soluble proteins[58,60]. The coral *Acropora acuminata*, for example, was shown to allocate up to 40% of its daily carbon fixation to mucus production, using it to trap and shed adhering

sediment[61,62]. It is therefore plausible that partial aerobic respiration was needed to meet the metabolic demand of mucus production in *P. periphylla*. This idea is further supported by a study investigating gene expression in mucus and tissue of the jellyfish *Aurelia coerulea* in search of bioactive compounds[33]. By removing *A. coerulea* from seawater, jellyfish started to produce excess mucus during which genes related to both aerobic and anaerobic metabolism were expressed in the jellyfish tissue (i.e. indicated by the KEGG pathways for oxidative phosphorylation, citrate cycle and glycolysis)[33]. Moreover, both mucus and tissue of *A. coerulea* showed over-expressed genes related to KEGG pathways involving ribosomes, phagosomes, lysosomes, focal adhesion, purine and pyrimidine metabolism, complement and coagulation cascades, and protein digestion or absorption[33]. These KEGG pathways were also observed in *P. periphylla*, and, although the stressors to which both jellyfishes were exposed differ, their overexpression supports metabolic investment in mucus secretion.

In addition to the excess mucus production, our results show a wide transcriptomic response of *P. periphylla* to suspended sediment. Similar to *P. periphylla* exposed to a four-degrees rise in temperature, expression of genes related to genetic information processing (e.g. ribosomal proteins) support their notion of being general regulators during environmental stress[42–44]. On the other hand, overexpressed genes similar to a zinc metalloproteinase, an anemone toxin, protease inhibitors or those related to protein degradation (e.g. proteolysis), and innate immunity could represent functions related to mucus production. Not only were similar genes and processes expressed in *A. coerulea* during mucus production[33], but mucus in cnidarians is well known to contain toxins and anti-microbial proteins that lyse and digest pathogenic cells[24,33,58,63]. Due to this function, protein cascades related to the breakdown of proteins may support the clearing of potential pathogenic molecules, while immune compounds could further help rid these unwanted molecules[47,58,64]. Although we did not directly observe any damaged tissues caused by the adhering sediment, expressed genes related to coagulation and wound repair may indicate tissue damage in *P. periphylla*[47]. Cnidarians are known for their ability to heal and sometimes even completely regenerate compromised tissues[44,47]. Genes related to wound repair, for instance, were also upregulated in the physically stressed coral *Paramuricea biscaya* in response to a deep water oil spill[43], and in the coastal coral *Gorgonia ventalina* injected with a single-celled parasite[44]. In addition, upregulated genes in *P. periphylla* related to epithelial morphogenesis and focal adhesion may further support the wound repair function, as these are all part of the cnidarian immune repertoire[47,48,52]. Although we can only speculate about the exact involvement of the latter pathways in the absence of similar studies, the wide transcriptomic response suggests that prolonged exposure to suspended sediment has the potential to alter metabolic pathways and health in *P. periphylla*. Whether medusae can recover after exposure remains subject of further investigation, making this study a starting point to unravel their underlying cellular stress response.

The impact of suspended sediment on microbial community composition in *P. periphylla* suggests tolerance towards short-term plume exposure. Surprisingly, *P. periphylla* exposed to the highest sediment concentration showed a significantly different microbiome from the other treatments in the first set of experiments, a result that could not be replicated in the second round of experiments. While this discrepancy may represent some natural daily variability of the associated microbiomes, it is likely that differences were in part caused by the accumulation of ASVs from the sediment onto the *P. periphylla* outer bell. Since *P. periphylla* produced large quantities of mucus that slowly sloughed off, it appears that medusae in the second round had cleared more sediment from their bodies at the end of our experiments compared to those in the first round. Moreover, once the sediment and associated bacteria were cleared (i.e. in the second round), *P. periphylla*'s microbiome was similar across all treatments. As such, our results instead support the protective function of mucus to ward off microbes and particles. Similar tolerances to suspended sediment were observed in the microbiomes of six species of coastal sponges, which were exposed to simulated sediment plumes ex situ for up to 28 days[65,66]. Nevertheless, we should note that we cannot rule out any long-term effects, as mining plumes may last well beyond the incubation periods currently studied[20–22] and *P. periphylla* may not be able to maintain the costly mucus production for prolonged periods of time.

Here, we present the first physiological, molecular and microbial response of a deep pelagic jellyfish to environmental stressors, taking a crucial step towards assessing the effects of stressors on deep pelagic ocean health. Future research should focus on the effects of a combination of stressors, as we could not include interactions in our experimental design due to space and logistical limitations aboard the ship. Interactions between the two stressors, for instance, have been observed in gorgonian corals, which reduced their metabolism as a response to sediment exposure, but increased both respiration and nitrogen excretion when sediment and warming were combined[67]. Moreover, many of the differentially expressed genes could not be annotated through sequence homology, highlighting the importance to study deep pelagic non-model organisms to gain further insights into their stress response and identify useful biomarkers for environmental surveys. In addition, we should emphasize that the exact concentrations of sediment plumes generated by a mining operation will be highly dependent on the initial amount of sediment discharged, the sediment type and hydrographic features[20–22]. Choosing the right concentrations of sediment exposure is therefore open to further improvement. In spite of this, it is worth noting that our sediment treatments not only mimic a range of distances from an initial sediment release site, the settling of sediment in our experimental tanks also represents in situ sedimentation and dilution of a plume and places the overall sediment exposure in the ranges of experimentally measured plume concentrations[20].

Our results show that *P. periphylla* can quickly respond to environmental stressors by invoking several physiological compensation strategies. Although our results point towards potential respiratory tolerance in *P. periphylla* towards deep ocean warming scenarios (1 °C over the next 84 years), it should be stressed that *P. periphylla*'s metabolic response to suspended sediment exceeds the most extreme warming treatment (4 °C). Not only did the highest sediment concentration lead to a doubling of respiration rates (similar to the four-degrees rise in temperature), but suspended sediment provoked an energetically costly response through the production of excess mucus under all sediment loads. Moreover, where the four-degrees rise in temperature is likely most energetically costly in the long run, short-term exposure to suspended sediment led to acute physical effects. Although the exact effects and metabolic costs will depend on suspended sediment concentrations, our results suggest that prolonged sediment exposure has the potential to offset energy demands that have taken millions of years to evolve[14], even at low concentrations. Specifically, if mining plumes are present in the water column for extended periods of time, elevated metabolism has either to be met with increased food intake, or sustained mucus production will lead to energetic depletion and thereby a reduction of health and possibly death. If *P. periphylla*'s response is representative of other gelatinous organisms, a diverse and abundant group of animals in the deep ocean[3], the commercial mining of the deep seafloor may impact biodiversity and the healthy functioning of the deep ocean, including its many services. The start of deep-sea mining is approaching at a rapid pace, and our results stress that caution should be taken in regulating mining activities to protect the largest, yet least explored habitat on the planet.

## Methods

### Sampling campaign

*Periphylla periphylla* were collected in the Lurefjord (60.692°N 5.155°E) and Sognefjord (61.100°N 5.585°E) between 2–19 March aboard the *R/V Heincke* (HE570) and 11–23 November aboard the *R/V Alkor* (AL568) in 2021, respectively. Sampling permits were obtained from the Norwegian Directorate of Fisheries, including permits 20/14555 (HE570) and 21/73103 (AL568), without needing ethical approval for experiments with jellyfish. To characterize their vertical distribution (Supplementary Fig. 7) and determine in situ temperature for the bulk of the population (Supplementary Fig. 8), oblique hauls with a Hydrobios© Multinet Maxi were conducted (0.5 m² in diameter, nine nets, 2 mm mesh, equipped with flowmeters). For experiments, individuals were caught between 0 and 800 m depth after sunset (i.e. for precise collection depths see the PANGAEA dataset https://doi.org/10.1594/PANGAEA.957367), using slow vertical hauls of conical plankton nets (either WP2 net, 57 cm diameter, 200 μm mesh or WP3 net, 113 cm diameter, 1000 μm mesh) with a non-filtering cod-end to minimize damage. Water for the experimental tanks was collected below the thermocline with Niskin bottles attached to a CTD rosette at 300 m depth, to avoid catching copepods that were present at shallower depths. Additional water samples were taken as a reference for the microbiome analysis at 12 depths across the entire water column (i.e. 10, 20, 40, 60, 80, 100, 150, 200, 250, 300, 300, 470 m) on 7 March 2021. For each depth, 2 L of water were filtered on MF-Millipore membrane (pore size 1 μm; diameter 25 mm). Since the red porphyrin pigment in *P. periphylla* is phototoxic[68], jellyfish were kept in the dark, and handling or manipulation of the experimental tanks was done under red light.

### Temperature and sediment plume experiments

Individuals that were undamaged after net sampling and therefore suitable for experiments were acclimatized for 7 to 14 hours in seawater at their average in situ temperature below the thermocline (7.5 °C; Supplementary Fig. 8). For the temperature incubations, this meant that jellyfish were kept in 60 L tanks, while those for the simulated plume experiments were kept in 30 L kreisel tanks (Schuran Seawater Equipment) with continuous water circulation maintained by gentle aeration from air pumps, introducing small bubbles on the side of the tanks (< 0.5 cm, ~1-2 per second), mid height, to avoid contact with the jellyfish.

To estimate the temperature-dependence of *P. periphylla*'s metabolic activity, jellyfish were exposed to one of three temperature treatments (7.5, 9.5 and 11.5 °C) lasting 7 to 9 hours. Since the *P. periphylla* from this study were obtained from coastal fjords and undergo shallow diel vertical migrations during winter, they are naturally exposed to temperature changes of 2–4 °C. Water temperatures were therefore not only chosen to mimic ocean warming scenarios (i.e. ~1 °C over the next 84 years, with 4 °C warming only under extreme climate predictions)[41], but also to estimate temperature-dependence of *P. periphylla* metabolic activity across equally increasing temperature treatments. Sample sizes for each temperature treatment were determined by the quantity and size of jellyfish caught, in addition to the availability of our respiration chambers, which occasionally meant splitting our treatments between days (i.e. see Supplementary Table 1 or the PANGAEA dataset https://doi.org/10.1594/PANGAEA.957367). At the start of the temperature incubations, individual medusa were transferred into respiration chambers (0.85, 1.15 or 2.5 L depending on jellyfish size) connected to fiber optic oxygen meters (PreSens Oxy–4 mini). Seawater in these chambers was pre-aerated to 100% air saturation and kept at respective temperatures overnight prior to the incubations. Rates of oxygen consumption were measured at the start and end of the incubation period and corrected for microbial respiration by including a control for each temperature run. To ensure no heat shock or handling effect of *P. periphylla* was

recorded in response to their transfer into respiration chambers, the first two to four hours of incubations were monitored. In addition to the oxygen consumption, ammonium excretion was determined for individuals in the temperature treatments as an end-point-measurement using the fluorometric method by Holmes et al.[69], using a Turner Trilogy fluorometer with a UV-module. Additional ammonium measurements for individuals that were too small for the respiration chambers (i.e. 2.16–2.35 cm CD) were done in 0.6 L air-sealed glass bottles with the same conditions as noted above.

### Sediment suspension

To determine the effect of sediment plumes on *P. periphylla*, jellyfish were exposed to five sediment concentrations (i.e. 0, 16.7, 33.3, 166.7, 333.3 mg·L⁻¹) at their average in situ temperature (7.5 °C) for 24 hours. Sediment concentrations were chosen to represent a range distances from a mining site, as plume dynamics will be highly dependent on the quantity of released sediment, sediment type and hydrographic features[20,21]. As such, the sediment treatments also represent a range of initial discharge concentrations, with the dilution and sedimentation of mining plumes mimicked by the settling of sediment in the experimental tanks (i.e. durations of in situ exposure unknown, with concentrations here decreasing 50% after 4-6 hours of incubation, Supplementary Fig. 10, measured with a Turb® 340 IR meter). Abyssal plain sediment was obtained in the North Atlantic (47.250°N 10.105°W) onboard the *R/V Sonne* using a box corer from 4427 m depth between 4 December 2020–5 January 2021 (SO279) and stored at -80 °C. Sediment consisted of organic aggregates (2 – 385 μm) and fine organic and inorganic particles (0.5–120 μm, e.g. diatom shells, sediment grains; Supplementary Fig. 9). Sample sizes for each experimental run were determined by the quantity and size of jellyfish caught to allow enough space for normal behavior, similar to that observed in individuals in situ[70], in each of the five experimental tanks. As such, kreisel tanks always contained up to four small (0.83–5.33 cm CD) or one large (6.40–10.54 cm CD) *P. periphylla*. Each of the five sediment concentrations was randomly assigned to kreisel tanks, resulting in roughly similar size distributions across treatments (i.e. on average ranging between 2.96 and 4.09 cm in CD, see Supplementary Table 1 or the PANGAEA dataset https://doi.org/10.1594/PANGAEA.957367). During the incubation period, the physical condition of *P. periphylla*, and water temperature (± 0.3 °C change over time) were monitored. Since we could not directly measure oxygen consumption in the kreisel tanks with oxygen optodes due to difficulties in sealing the tanks air-tightly and given the large tank volume relative to the animals, electron transfer system (ETS) activity was analyzed as outlined in the Supplementary Materials to act as a proxy for respiration rate. The physical condition of jellyfish was scored based on the amount of mucus produced and sediment aggregated on the jellyfish's body. Final scores ranged between 1 (worst) to 4 (best) and were assigned to individuals every 6 h during the 24h-time period (i.e. at 0 h, 6 h, 12, 18 h, 24 h). The criteria for the respective scores were 1: >30% sediment-mucus coverage, substantial sloughing of mucus (i.e. on bell, tentacles and edge of the bell also known as lappets), 2: <30% sediment-mucus coverage, moderate sloughing of mucus (i.e. on lappets only), 3: Powder-like mucous cover, no mucus sloughing, 4: Clean bell (Fig. 4). Temperature of all experimental tanks was monitored throughout the experiments.

Directly after all experiments, *P. periphylla* were photographed (Olympus TG-6 camera) for later size determination. To limit the effect of handling on jellyfish metabolism, whole jellyfish were first snap frozen at -20 °C to minimize warming of the -80 °C freezer and then transferred to -80 °C within 5 minutes of being taken from their experimental containers. Coronal diameter (CD) and height (CH) were measured from photographs with ImageJ v1.52k[71] to estimate wet weight (WW) following equations by Fosså[36]. Tissue samples for ETS, microbiome and gene expression (transcriptome) analysis were taken

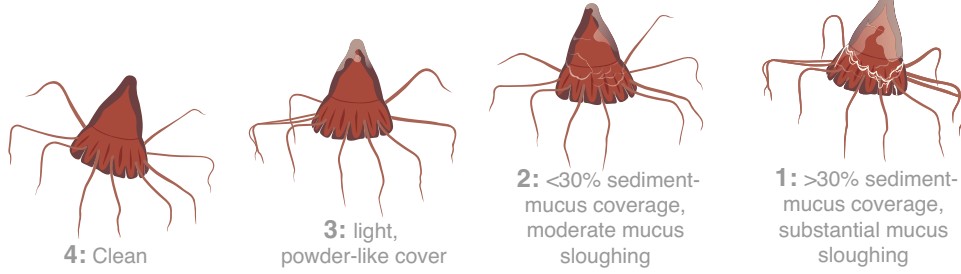

**2:** <30% sediment-mucus coverage, moderate mucus sloughing

**1:** >30% sediment-mucus coverage, substantial mucus sloughing

**3:** light, powder-like cover

**4:** Clean

**Fig. 4 | Illustrated health rating based on the amount of sediment aggregated on the bell and mucus produced.** Scores indicate: 1. >30% sediment-mucus coverage, substantial sloughing of mucus (i.e. on bell, lappets and tentacles), 2. <30% sediment-mucus coverage, moderate sloughing of mucus (i.e. on lappets only), 3. Powder-like mucous cover, no mucus sloughing, 4. Clean bell.

from frozen jellyfish in the laboratory back on shore. Only samples from the HE570 cruise were used for microbiome and transcriptome analysis due to project timing constraints.

## Transcriptome analysis

For the transcriptome analysis, RNA was extracted from jellyfish as described in the Supplementary Materials. RNA concentration was measured on a NanoDrop® spectrophotometer, while the quality of RNA extracts was tested using gel electrophoresis on an Agilent Bioanalyzer. Samples with RIN values of >7 were used for library preparation and de novo transcriptomic sequencing on the DNBSeq platform (150 bp paired-end reads) at BGI Tech Solutions Co., Hong Kong, yielding ~30 M reads. Quality check and filtering of raw reads was carried out by BGI, removing adapter sequences, contamination and low-quality reads (i.e. output read quality set to Phred score +33) with the SOAPnuke[72] software.

Since no reference genome is available for *P. periphylla*, RNA reads were assembled into a de novo transcriptome using Trinity v2.8.5[73] with default parameters in paired-end mode. The generated transcriptome was trimmed using Transdecoder v5.50[73] by selecting open-reading frames (ORFs; at least 100 bp in length) and predicting candidate coding regions. Transcriptome annotation was performed with Trinotate v3.2.1 searching the Blastx, Blastp, Pfam, UniProt-SwissProt, GO, eggNOG and KEGG databases[74], followed by removal of possible contamination matching sequences only known from prokaryotic or viral origins (blastx, e-value threshold 1e$^{-5}$). Read alignment and abundance estimation were performed in Trinity using the Bowtie2 2.5.0[75] and Salmon 0.14.1[76] plugins respectively. To ensure no important transcripts were lost in the trimming process, transcriptome composition for both the trimmed and full transcriptome was assessed with BUSCO (Benchmarking Universal Single-Copy Orthologs)[77] using the metazoan OrthoDB v09 reference database (see Supplementary Table 1 for assembly statistics). Removal of contamination was further confirmed by the overall sequence GC content, which was slightly higher after trimming (Supplementary Table 2).

To identify differentially expressed (DE) genes, sediment, and temperature experiments were analyzed separately, comparing changes in expression across treatments with the likelihood ratio test (LRT). Since *P. periphylla* in the 33 mg L$^{-1}$ treatment consisted of a single replicate, and we could not reliably assess the effect of treatment versus biological variability, this treatment was not included in downstream analysis. Normalized expression matrices were constructed with DESeq2[78] in R v4.2.1[79] using the abundance estimates generated by Salmon in quasi-mapping mode. Transcripts were further trimmed by removing those with zero normalized counts across all samples, reducing the de novo transcriptome to 293,318 transcripts. To further eliminate noise within the matrices and obtain the final transcriptome count, normalized counts were filtered to only include transcripts with a count of at least ten in two or more samples. To visually compare transcriptomes across individuals and treatments,

principal component analyses (PCA) were carried out on log2 transformed counts with the R package factoextra[80]. Tested models within the DESeq2 analysis included size + date + treatment for the suspended sediment experiments, and size + treatment for the temperature experiments, leaving out date in the latter as not all temperature treatments were run on the same experimental days resulting in non-fully rank model. To implement the LRT in DESeq2, the full models mentioned above were tested against reduced null models to assess the effect of the full models and identify gradients of up- and down-regulated genes with our ranges of increasing sediment concentrations and temperatures. Genes were considered significantly expressed if adjusted p-values (p-adj.) were below 0.05 (using the Benjamin-Hochberg false discovery rate; FDR) with a log2 fold change (lfc) threshold value of > 0 for over expressed genes and <0 for under expressed genes. Expression patterns were identified and visualized using the R package DEGreport, using log2 transformed expression counts[81]. Moreover, heatmaps of all significantly over-expressed genes were constructed using the pheatmap package in R, transforming expression values as z-scores (i.e. subtracting average gene abundance as calculated from all samples, divided by the standard deviation).

The biological function of significant DE genes (p-adj. <0.05) was assessed using the Kyoto Encyclopedia of Genes and Genomes (KEGG) mapper reconstruct tool v5.0[38], based on KEGG Orthology (KO) numbers assigned through eggNOG v2.1.9[82]. To further describe the biological role of significant DE genes, we implemented a gene ontology (GO) enrichment analysis using the R based package ClusterProfiler[83,84], focusing on biological processes for those groups that showed an increase or decrease across treatments.

## Microbiome analysis

Microbial DNA was extracted and sequenced from jellyfish epithelial tissue, seawater and abyssal sediment as described in the Supplementary Materials. The bioinformatic pipeline published by Busch et al.[85] was followed for analysis of generated sequences using R v3.5.1[79] and QIIME2 (v2018.11 unless otherwise specified)[86]. Following removal of primer sequences, quality check of raw reads was performed in QIIME2. Forwards reads were used to generate Amplicon Sequence Variants (ASVs), which were trimmed to 275 bp, removing singletons and chimeric sequences, using the DADA2 algorithm[87]. Taxonomy of ASVs was identified with QIIME2 v2019.10 using a Naïve Bayes classifier trained on the Silva 132 99% OTUs 16 S rRNA database[88]. Chloroplasts and mitochondrial sequences were removed from the dataset, followed by subsampling of the data to a sampling depth of 6700 reads per sample. The FastTree2[89] plugin was used to estimate a phylogenetic tree and core-diversity metrics were calculated (i.e. Shannon index and weighed UniFrac distances)[90]. Again, *P. periphylla* in the 33 mg L$^{-1}$ treatment were omitted from further analysis as only one replicate was available within the HE570 dataset. To visually compare microbiomes across individuals and treatments, non-metric multidimensional scaling (NDMS) was performed on weighed UniFrac

distances. Finally, the R based package UpSetR[91] was used to identify shared microbial taxa among treatments and whether bacteria were transferred from the abyssal plain sediment to *P. periphylla* in the sediment treatments.

## Statistics and reproducibility

To assess statistical differences between respiration rates, ammonium excretion and the health score rankings (testing both treatment and time point), we implemented one-way ANOVA tests followed by a post-hoc Tukey test for significant results. For each analysis, the normality of residuals and homogeneity of variances were assessed using the Shapiro-Wilk and Levene's tests, respectively. To investigate differences among microbiome samples, comparing effects of tissue type (i.e. water, sediment, jellyfish), treatment, jellyfish size, and date, we implemented permutational multivariate analyses of variance (PERMANOVA) with 999 permutations. As mentioned above, significant differentially expressed genes were identified using the DESeq2 package based on the Likelihood Ratio Test. Data and metadata for *P. periphylla*'s physiological measurements can be found on the PANGAEA repository (https://doi.org/10.1594/PANGAEA.957367), in addition to the microbial community composition (https://doi.org/10.1594/PANGAEA.957395) and the normalized expression counts for differentially expressed transcripts (i.e. including lfc values of significantly DE transcripts and annotations; https://doi.org/10.1594/PANGAEA.962217). Transcriptome and microbiome sequences are hosted on NCBI's SRA database (BioProject ID: PRJNA971902 with accession numbers SAMN35056874-902 and PRJNA971258 with accession numbers SAMN35028032-079, respectively). Experiments for the temperature treatments were repeated twice during the HE570 and AL658 cruises, while the sediment treatments were repeated twice during the HE570 and AL568 cruises each (for a total of four times). Detailed information on the timing and repetition of experimental runs is available on PANGAEA in the metadataset (https://doi.org/10.1594/PANGAEA.957367).

## Reporting summary

Further information on research design is available in the Nature Portfolio Reporting Summary linked to this article.

# Data availability

Data for *P. periphylla*'s physiological measurements has been deposited in the PANGAEA repository (https://doi.org/10.1594/PANGAEA.957367), in addition to the microbial community composition (https://doi.org/10.1594/PANGAEA.957395) and output from the transcriptome analysis (i.e. normalized counts, lfc values of significantly expressed genes and annotations for expressed transcripts; https://doi.org/10.1594/PANGAEA.962217). Transcriptome and microbiome sequences are hosted on NCBI's SRA database under the BioProject IDs PRJNA971902 with accession numbers SAMN35056874-902 and PRJNA971258 with accession numbers SAMN35028032-079, respectively. The assembled transcriptome is available on Figshare (https://doi.org/10.6084/m9.figshare.24114369.v1) with the BUSCO metazoan OrthoDB v09 database available on https://busco-data.ezlab.org/v5/data/lineages. The data presented in figures, underlying means, box plots and scatter plots, are available in the accompanying Source Data file. Source data are provided with this paper.

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

## Acknowledgements

We are grateful to the crew of *R/V Heincke* HE570 and *R/V Alkor* AL568 for their support during the sampling campaign. We thank Aaron Beck for collection of the sediment samples, and Maya Bode for sharing her knowledge for the ETS analysis. We thank Ina Clefsen for the DNA extractions of the microbiome and Andrea Eschbach for teaching VIS the RNA extraction protocol. This study received funding from the European Union's Horizon 2020 research and innovation program under grant agreement no. 818123 (iAtlantic). This output reflects only the authors' view and the European Union cannot be held responsible for any use that may be made of the information contained therein. HJH received funding from the Deutsche Forschungsgemeinschaft (DFG) under grant HO 5569/2-1, an Emmy Noether Junior Research Group awarded to HJH. CH is funded by the Helmholtz Young Investigator Group "ARJEL – Arctic Jellies" with the project number VH-NG-1400, funded by the Helmholtz Society and the Alfred Wegener Institute Helmholtz Centre for Polar and Marine Research.

## Author contributions

H.J.H., H.H., and A.K.S. applied for funding, while H.J.H. and H.H. conceived and managed the project. A.K.S. provided part of experimental equipment. H.H. and V.I.S. carried out all field work, experiments, and lab measurements of $O_2$ consumption, $NH_4$ production, and ETS activity. C.H. and their team developed the RNA extraction protocol, while U.H. and their team developed and performed the microbiome 16 S rRNA extraction protocol. L.S. and T.B. advised V.I.S. on the microbiome and transcriptome analysis, respectively. V.I.S. performed the RNA extractions, curated and analyzed all data, and prepared all figures. V.I.S. wrote the manuscript, which was reviewed and approved by all authors.

## Funding

## Competing interests

The authors declare no competing interests.
