## [Peer Review File · Nature Communications]

Experimental mining plumes and ocean warming trigger stress
in a deep pelagic jellyfishREVIEWER COMMENTS

Reviewer #1 (Remarks to the Author):

The paper by Stenvers et al is a first look at the potential effects of deep-sea mining plumes on the biology of a pelagic animal. They also look at warming as a stressor for comparison and here there is more data in the literature (though little on jellies). The authors have a very nice multifaceted data set including respiration experiments, microbiome analysis and genetic expression. The paper is very well written and is incredibly important to publish as mining regulations are being evaluated right now and several groups are trying to determine thresholds for sediment exposure to deep-sea animals.

That said, *P. periphylla*, while a great animal to work with is probably not "representative" of many pelagic taxa (see below). Further, while the study presents great results they are for one species so the interest in this study to a broad audience may be somewhat limited. This is a case study, even if the first.

Specific comments

1) The authors need to pay more attention to sediment concentration throughout. The sediment concentration should be specified in the abstract just as the number of degrees of warming was specified. The discussion of the sediment effects (line 404 onwards) largely ignores concentration dependant effects. Suspended sediment loads of 333 mg/l are VERY high and likely occur only near the sites of sediment discharge. Much lower concentrations will occur over much larger areas and the plume dimensions cited earlier in the paper are for a return to background which is about 0.02mg/l in the CCZ, for instance. The authors should discuss the 17.6 mg/l treatment results in particular. These concentrations are still quite high for open ocean environments. How do the effects observed at this concentration (which appear significant from fig2a,b) inform the possible ecological consequences of deep sea mining?

2) line 64 - I realize the authors are making a general statement about thermal tolerances in pelagic animals but diel vertical migrators daily encounter rather pronounced thermal changes that are well adapted to and indeed are the subject of quite a bit of literature (e.g.

Seibel and Birk, 2022, that the authors cite a couple of sentences further on). Given that the animals in this study are derived from the mesopelagic a bit more detail about thermal tolerances in the introduction is warranted.

3) line 84-85 - the authors cite Munoz-Royo et al about sediment plumes that suggest plumes may extend a 10's to 100 km not 100's. That is still a very large extent but is a better supported statement. Revise the sentence.

4) The authors argue that *P. periphylla* is a representative deep pelagic animal because it is cosmopolitan and eurybathic. However, these traits make this species unusual as there are few true cosmopolitan species in the pelagic and even less species are eurybathic over such an impressive depth range. I would argue that studying this animal is ideal because it can be captured shallower but also occurs deeper and while found in Norwegian fjords, studies of its physiology and biology are relevant to populations found elsewhere, where mining is likely to occur (e.g. the CCZ of the Pacific). This should be made clear.

5) The figures are really well done with nice graphics to orient the reader. A very minor point but consider making the type on some of the panels (e.g. Fig 1,2 d and e) a bit bigger so its easier to read.

6) line 148 - the authors state respiration was measured from "fiber optics" but the salient point is that it was measured as "oxygen consumption" please change

7) line 200-202 - Is the significant effect of date for the microbiomes only or also for the corresponding seawater samples? please specify. See also discussion line 399.

8) line 305 and line 307 - Please clarify what "in 3 out of 4 individuals" refers to. Is this the number of jelly individuals or the proportion of ASVs transferred from sediment to the jelly that were present in a particular treatment?

9) paragraph starting line 333 - Much of the authors discussion of the why there was a weak increase in metabolic rate (O₂ consumption) is predicated upon a paper suggesting this

species can undertake both aerobic and anaerobic pathways simultaneously (Thuesen and Childress paper). The authors go so far as to suggest that the jellies increase anaerobic metabolism in response to temperature (line 337). The authors own results find no DE genes related to anaerobic metabolism though a weak and non-quantifiable shift away from those involved in aerobic metabolism. An equally plausible explanation is that the authors lacked the statistical power to see statistically significant increases in O₂ consumption. Their data is limited (n=3 at 11.5 C) and variable (see fig 1a). This is not a complaint! This data is hard to acquire and often variable. Further, their ammonium excretion data shows strong increases as should occur with increases in aerobic catabolism of proteins not anaerobic catabolism of carbohydrates. In summary, the authors should greatly temper their discussion of anaerobic metabolism which is only hinted at by their molecular data.

10) line 516-518 - The authors should specify WHEN the water samples were taken for microbe profiling and comparison to the jelly's microbiomes because they found a significant difference between times of collections for the jellies. Ideally they need water samples from each time period and make these comparisons (jelly to seawater) only at the same times.

11) line 528 - Can the authors specify if the "air pumps" actually introduced air bubbles into the experiments and exposed the jellies to another unnatural stressor! I doubt it but it should be specified to remove all doubt.

12) line 559 - Its not clear if the inability to measure o₂ consumption in the kreisels had to do with sealing them (which is hard!) or from the added effects of sediment. Can the authors specify please?

13) supplemental methods "V. hombuffer" is not a term given in the ETS equation but is defined. Please explain why.

14) supplemental methods. The last sentence of the ETS section suggests a coefficient to convert from ETS activity to oxygen consumption. This data, including sample size and regression statistics, should be presented and shown because its critical to interpretation of

the data, particularly in comparing temperature effects on metabolic rates (measured directly as oxygen consumption) to sediment effects (measured as ETS activity).

Reviewer #2 (Remarks to the Author):

This research presents information on experimental exposure of a common pelagic jellyfish to a range of temperatures and sediment regimes that are designed to represent future ocean warming scenarios and sediment plume exposure from deep sea mining. The experiments for this research were conducted on board the research vessel and were not ideally replicated; however, the work is novel and innovative, particularly with regard to genetic responses, and provides valuable insight into the impacts of anthropogenic stressors on a dominant pelagic species. The authors' conclusions support the research results as far as I can tell, although I am not an expert in genomics so the editor should defer to other reviewers on this point. The methodology seems appropriate, with the caveat that it was conducted under non ideal conditions, but there are some gaps in description of experimental design that need to be filled to help the reader fully understand what was done. In summary, my recommendation is publish with minor revisions. I have included comments and questions in the annotated manuscript.

Reviewer #3 (Remarks to the Author):

I was asked to review the omics approaches detailed in the manuscript. Overall, the methodology employed is commonly used state of the art for the RNAseq and Microbiome pipeline. However, magnitude of molecular responses observed makes me question the interpretation of the data; the authors state 'worrisome implications' when in fact gene expression and microbiome response are marginal (physiological data set aside).

Sequencing data is not available on NCBI under the provided BioProject IDs; these data must be made available prior to any publication. Also, it is nowadays fairly common to provide the bioinformatic workflow, e.g. via GitHub as well as the assembled transcriptome. Both are not made available [This is even more curious, since physiological and microbiome data are provided]. Thus, the data produced in the manuscript is of limited use to other

researchers.

I was pretty surprised to see the very high number of assembled transcripts without any discussion. At large, it is rather hard to follow how things were done with how many replicates and what cutoffs. A flow-chart as a supplemental figure and a supplemental table detailing the numbers would be a huge help.

I have to say that I find the differential gene response marginal for the temperature treatment (167 up and 54 down-regulated out of 20,697 putative genes < 1%; the change from 839,298 transcripts to 20,697 is also not detailed/documented/discussed). Is this tiny response expected? what is the average fold change? Why is it stated 'genes' here when transcripts were assembled? I also find the up regulation of metabolism under increased temperature less than surprising .. this relates pretty much to any organism.

Response to reviewer comments

We are grateful to the reviewers for their constructive criticism. In the following, we provide point-by-point responses (in blue) detailing how we addressed the feedback by the reviewers. Line numbers refer to the cleaned and final version of the manuscript.

The most important revisions include text improvements, including additional clarification that we investigated simulated and not *in situ* sediment plumes, more detailed explanation of the methods and the trimming of the Discussion paragraph discussing aerobic and anaerobic metabolism in response to temperature.

Reviewer #1

The paper by Stenvers et al is a first look at the potential effects of deep-sea mining plumes on the biology of a pelagic animal. They also look at warming as a stressor for comparison and here there is more data in the literature (though little on jellies). The authors have a very nice multifaceted data set including respiration experiments, microbiome analysis and genetic expression. The paper is very well written and is incredibly important to publish as mining regulations are being evaluated right now and several groups are trying to determine thresholds for sediment exposure to deep-sea animals. That said, *P. periphylla*, while a great animal to work with, is probably not "representative" of many pelagic taxa (see below). Further, while the study presents great results they are for one species so the interest in this study to a broad audience may be somewhat limited. This is a case study, even if the first.

Thank you very much, we highly value your feedback on our manuscript. While we agree that *P. periphylla* may not be representative for all pelagic taxa, we would like to emphasize that it does represent the great diversity and abundance of pelagic animals belonging to the phylum Cnidaria and, perhaps more importantly, is a globally distributed deep sea species. Since deep-seabed mining is potentially taking place on a global scale, it was therefore important to study an organism that occurs in multiple habitats and regions. Currently, a study comprising multiple deep pelagic taxa seems unfortunately not feasible given the difficulty in studying deep-water animals, in particular in an experimental setting, and we acknowledge the limitations of our study. However, we do hope our study spurs follow-up experimental and observational work on pelagic organisms. As such, future directions and limitations of our study are highlighted in the second to last paragraph of the discussion. Given the current discussion on the commercial mining of the deep seabed, the absence of similar studies investigating mining effects on deep pelagic ecosystems, and our holistic approach (combining physiology, gene expression and microbiomes, while presenting the first transcriptome and microbiome for a deep pelagic jellyfish), we believe our results are of interest to a broad readership with Nature Communications as a suitable outlet for our manuscript.

Specific comments

1) The authors need to pay more attention to sediment concentration throughout. The sediment concentration should be specified in the abstract just as the number of degrees of warming was specified. The discussion of the sediment effects (line 404 onwards) largely ignores concentration dependent effects.

Suspended sediment loads of 333 mg/l are VERY high and likely occur only near the sites of sediment discharge. Much lower concentrations will occur over much larger areas and the plume dimensions cited earlier in the paper are for a return to background which is about 0.02mg/l in the CCZ, for instance. The authors should discuss the 17.6 mg/l treatment results in particular. These concentrations are still quite high for open ocean environments. How do the effects observed at this concentration (which appear significant from fig2a,b) inform the possible ecological consequences of deep sea mining?

Thank you for your suggestion, the sediment concentration was added to the abstract (L43). Although we suspended a given amount of sediment in the tanks, the concentration did decrease over time due to sedimentation. Therefore, the animals were not exposed continuously to the nominal concentration, but a gradually and substantially lower one. Nominal concentrations in the higher treatments are comparable to those near the initial sediment discharge (Muñoz-Royo et al. 2021. *Communications Earth & Environment*), but the exposure is more in the range of the actual plume. To address the effect of the sediment dilution, we have added the following to the Discussion and Methods:

(L758 Discussion): ‘In addition, we should emphasize that the exact concentrations of sediment plumes generated by a mining operation will be highly dependent on the initial amount of sediment discharged, the sediment type and hydrographic features²⁰⁻²². Choosing the right concentrations of sediment exposure is therefore open to further improvement. In spite of this, it is worth noting that our sediment treatments not only mimic a range of distances from an initial sediment release site, the settling of sediment in our experimental tanks also represents *in situ* sedimentation and dilution of a plume and places the overall sediment exposure in the ranges of experimentally measured plume concentrations²⁰.’

(L878 Methods): ‘As such, the sediment treatments also represent a range of initial discharge concentrations, with the dilution and sedimentation of mining plumes mimicked by the settling of sediment in the experimental tanks (i.e. decreasing 50% of total after 4-6 hours of incubation, Fig. S10, measured with a Turb® 340 IR meter).’

2) line 64 - I realize the authors are making a general statement about thermal tolerances in pelagic animals but diel vertical migrators daily encounter rather pronounced thermal changes that are well adapted to and indeed are the subject of quite a bit of literature (e.g. Seibel and Birk, 2022, that the authors cite a couple of sentences further on). Given that the animals in this study are derived from the mesopelagic a bit more detail about thermal tolerances in the introduction is warranted.

To better clarify, we have added the following to the Introduction:

(L66) ‘The exception here are organisms that are adapted to undergo vertical migrations, which may be less sensitive to changes in temperature than non-migrant species¹³.’

3) line 84-85 - the authors cite Munoz-Royo et al about sediment plumes that suggest plumes may extend a 10's to 100 km not 100's. That is still a very large extent but is a better supported statement. Revise the sentence.

Adjusted as follows:

(L90) ‘... the generated sediment plumes can affect the entire water column, expand for hundred kilometers and last for several years.’

4) The authors argue that *P. periphylla* is a representative deep pelagic animal because it is cosmopolitan and eurybathic. However, these traits make this species unusual as there are few true cosmopolitan species in the pelagic and even less species are eurybathic over such an impressive depth range. I would argue that studying this animal is ideal because it can be captured shallower but also occurs deeper and while found in Norwegian fjords, studies of its physiology and biology are relevant to populations found elsewhere, where mining is likely to occur (e.g. the CCZ of the Pacific). This should be made clear.

While we agree with the reviewer’s comment regarding the suitability of *P. periphylla* for experimental work, we would like to point out that meso- and bathypelagic species often have a wider distribution both geographically and vertically than epipelagic ones, which can be explained by the more narrow and stable range in environmental conditions. Since we have already noted that *P. periphylla* is a representative organism to study warming and mining as it represents global deep ecosystems (i.e. where mining is likely to occur), we have made the following changes to include the relevance for their biology:

(L139) ‘Moreover, *P. periphylla* is known for its high abundance in several Norwegian fjords, allowing for relatively easy and gentle collection of individuals while their biology and physiology remain relevant for oceanic populations^{36,37}.’

5) The figures are really well done with nice graphics to orient the reader. A very minor point but consider making the type on some of the panels (e.g. Fig 1,2 d and e) a bit bigger so it’s easier to read.

Thank you, we appreciate your feedback. We have made the fonts of panels d and e bigger in both figures as suggested.

6) line 148 - the authors state respiration was measured from "fiber optics" but the salient point is that it was measured as "oxygen consumption" please change

Adjusted to ‘oxygen optodes’ here (L176), and in the methods (L895).

7) line 200-202 - Is the significant effect of date for the microbiomes only or also for the corresponding seawater samples? please specify. See also discussion line 399.

A similar point regarding collection dates was raised by reviewer 2. To better clarify differences caused by collection date, we have changed the section as follows:

(L266) ‘When comparing the microbial community structure of *P. periphylla* across treatments, jellyfish microbiomes were significantly different from the sea water (PERMANOVA $df=1$, $F=23.20$, $p=0.001$, $n=24$) with no effect of the temperature treatments (Supplementary Fig. 4a, Supplementary Table 4). Nevertheless, the sampling date proved to significantly affect microbial composition (PERMANOVA $df=1$, $F=8.28$, $p=0.004$, $n=12$) since not all *P. periphylla* were collected on the same date due to logistic limitations aboard the research vessel. Consequently, microbiomes of individuals exposed to 9.5°C could

not be compared to those at 7.5 and 11.5°C, as individuals in the former treatment were collected on a different day.'

8) line 305 and line 307 - Please clarify what "in 3 out of 4 individuals" refers to. Is this the number of jelly individuals or the proportion of ASVs transferred from sediment to the jelly that were present in a particular treatment?

We meant the number of individuals in this particular treatment and have made the following amendment to better clarify:

(L465) 'Of these ASVs, 17 were recovered in *P. periphylla* exposed to 333 mg L⁻¹ of suspended sediment (i.e. in three out of four individuals in this treatment group), ...'

9) paragraph starting line 333 - Much of the authors discussion of the why there was a weak increase in metabolic rate (O₂ consumption) is predicated upon a paper suggesting this species can undertake both aerobic and anaerobic pathways simultaneously (Thuesen and Childress paper). The authors go so far as to suggest that the jellies increase anaerobic metabolism in response to temperature (line 337). The authors own results find no DE genes related to anaerobic metabolism though a weak and non-quantifiable shift away from those involved in aerobic metabolism. An equally plausible explanation is that the authors lacked the statistical power to see statistically significant increases in O₂ consumption. Their data is limited (n=3 at 11.5 C) and variable (see fig 1a). This is not a complaint! This data is hard to acquire and often variable. Further, their ammonium excretion data shows strong increases as should occur with increases in aerobic catabolism of proteins not anaerobic catabolism of carbohydrates. In summary, the authors should greatly temper their discussion of anaerobic metabolism which is only hinted at by their molecular data.

Thank you for your suggestion and thoughts on the matter. A similar point was raised by reviewer 2 and we agree that the manuscript will benefit by better highlighting the natural variability of oxygen consumption data (as mentioned in Thuesen and Childress, 1994. *Biological Bulletin*, and Youngbluth and Båmstedt, 2001. *Hydrobiologica*) and reflecting this in our discussion of aerobic respiration. To this end, we have made the following changes to the paragraph:

(L507) 'This moderate response in oxygen consumption can potentially be explained by *P. periphylla*'s ability to perform both aerobic and anaerobic respiration simultaneously³⁹, with downregulated genes here suggesting a shift away from aerobic metabolism. Although we did not find any significantly DE genes directly related to anaerobic metabolism (e.g. involving glycolysis), anaerobic respiration may have been continually active in *P. periphylla* across treatments. Consistent levels of anaerobic enzymes were also found in *P. periphylla* populations living at low versus higher oxygen concentrations, while those in low oxygen environments carried significantly lower levels of aerobic enzymes³⁹. Moreover, *P. periphylla*'s ability to respire anaerobically may contribute to the overall low respiration rates, which were within the range of previously reported values for *P. periphylla* and similar to those of other deep pelagic medusa capable of anaerobic respiration (Supplementary Table 7)^{35,40}. Nevertheless, further research is required to determine whether oxygen concentrations and seawater temperature have similar effects on aerobic versus anaerobic metabolism in *P. periphylla*. Alternatively, the response in oxygen consumption may still become significant when sampling sizes are increased, considering the natural variability in these animals

and associated data^{35,40}. However, based on our current results and given that *P. periphylla* can vertically migrate to significantly warmer waters at night and has a wide geographic distribution^{34,35}, we expect a relatively wide temperature tolerance in this species. Moreover, when considering current climate warming projections (i.e. approximately 1°C over the next 84 years, with 4°C warming only under extreme climate predictions)⁴¹, global warming scenarios for the deep ocean do not appear to pose an immediate respiratory threat for *P. periphylla*.’

10) line 516-518 - The authors should specify WHEN the water samples were taken for microbe profiling and comparison to the jelly's microbiomes because they found a significant difference between times of collections for the jellies. Ideally they need water samples from each time period and make these comparisons (jelly to seawater) only at the same times.

The date of water collection was added to the methods (L826). In hindsight, we agree that water samples for microbial analysis should have ideally been collected throughout the study period to align with jellyfish-collection days. However, we hope that the reviewer can appreciate our effort to show that timing of sampling is highly important for future microbiome studies of *P. periphylla*. Regardless of date, we are confident that we sequenced *P. periphylla*'s microbiomes in our jellyfish samples, as these are distinctively different to and considerably smaller than the microbial community in the seawater. Such select microbiomes are characteristic of jellyfish, which have bottom-up control over their microbiome through the expression of mucus and anti-microbial proteins (i.e. reviewed by Tinta et al. 2019. *Marine Drugs*).

11) line 528 - Can the authors specify if the "air pumps" actually introduced air bubbles into the experiments and exposed the jellies to another unnatural stressor! I doubt it but it should be specified to remove all doubt.

The bubbles were introduced at the side of the kreisel tank, mid height. The bubbles traveled along the inner curved wall of the tank to the surface in a steady stream of small (less than about 5mm) bubbles, approximately 1-2 bubbles per second. We did not notice an interaction between the bubbles and the jellies. Although kreisel tanks are the most suitable aquarium tanks for jellies, they are undoubtedly a very artificial environment (and thus likely stressful compared to *in situ*), which also limited the duration of our experiments. Since all tanks were the same, and aeration tuned to the same rate, we cannot test for the size of this unknown stressor "enclosure/aeration" but did limit any effects of it in our results through our control treatments. To better clarify, we have added the following:

(L837) ‘... with continuous water circulation maintained by gentle aeration from air pumps, introducing small bubbles on the side of the tanks (<0.5 cm, ~1-2 per second), mid height, to avoid contact with the jellyfish.’

12) line 559 - It's not clear if the inability to measure o2 consumption in the kreisels had to do with sealing them (which is hard!) or from the added effects of sediment. Can the authors specify please?

The main reason was the volume of the tank compared to the size of the organisms and the difficulty in sealing them air-tightly. That said, if both were possible to overcome, we think that effects of suspended

sediment could be controlled for by incubating controls with each sediment concentration but no animals. To better reflect this, the following changes were made:

(L894) 'Since we could not directly measure oxygen consumption in the kreisel tanks with oxygen optodes due to difficulties in sealing tanks air-tightly and given the large tank volume relative to the animals, electron transfer system (ETS) activity was analyzed as outlined in the Supplementary Materials to act as a proxy for respiration rate.'

13) supplemental methods "V. hambuffer" is not a term given in the ETS equation but is defined. Please explain why.

Thank you for pointing this out, the term was accidentally deleted from the equation (i.e. most right term) and should have appeared in front of ' $(\mu\text{L})/\text{WW}(\text{g})$ '. We have added the term back into the equation.

14) supplemental methods. The last sentence of the ETS section suggests a coefficient to convert from ETS activity to oxygen consumption. This data, including sample size and regression statistics, should be presented and shown because it's critical to interpretation of the data, particularly in comparing temperature effects on metabolic rates (measured directly as oxygen consumption) to sediment effects (measured as ETS activity).

Because of the low sample size, the regression fitted though the ETS activity vs. direct oxygen consumption (we could only use the temperature treatments for that) was not significant. We therefore used the mean ETS:R ratio to determine the conversion coefficient. All data has been uploaded to the PANGRAEA repository and will be publicly available with the publication of the manuscript. The link to the data has been added to the Supplementary Methods.

Reviewer #2

This research presents information on experimental exposure of a common pelagic jellyfish to a range of temperatures and sediment regimes that are designed to represent future ocean warming scenarios and sediment plume exposure from deep sea mining. The experiments for this research were conducted on board the research vessel and were not ideally replicated; however, the work is novel and innovative, particularly with regard to genetic responses, and provides valuable insight into the impacts of anthropogenic stressors on a dominant pelagic species. The authors' conclusions support the research results as far as I can tell, although I am not an expert in genomics so the editor should defer to other reviewers on this point. The methodology seems appropriate, with the caveat that it was conducted under non ideal conditions, but there are some gaps in description of experimental design that need to be filled to help the reader fully understand what was done. In summary, my recommendation is publish with minor revisions. I have included comments and questions in the annotated manuscript.

Thank you for your detailed review of our manuscript, we greatly appreciate your helpful feedback. We agree that we ideally would have liked more replications, and appreciate the recognition of valuable insights gained from the current dataset. Our sample sizes were chosen based on the number and capacity of experimental tanks to give the animals enough space for normal behavior, in addition to being determined by our overall ship-time on calm seas. Since we can support our findings holistically by approaching the stress-response from multiple angles (i.e. physical condition, respiration, ammonium excretion, gene expression and analysis of microbial symbionts), we have high confidence in our results, with similar sample sizes have previously been reported for transcriptomics and analysis of microbiomes. In spite of this, we were careful with our interpretations of the data, and to address any gaps in our methodology we have made several amendments as described in more detail below.

Page: 2

L38: The study did not directly assess the effects of global warming and sediment plumes. The experiments assessed responses to a range of temperatures and sediment concentrations which were chosen to represent a subset of conditions that may occur under future warming and mining scenarios. This may seem a trivial distinction but the use of this terminology throughout the paper is misleading. and should be amended. Here in the abstract, the addition of 'potential' before 'effects' would suffice.

We share the opinion that papers should avoid any misleading terminology and had hoped that describing our dataset as 'experimental' or 'simulated', in addition to naming our paper 'Experimental mining plumes and ocean warming trigger stress in a deep pelagic jellyfish', would resolve any ambiguity. As suggested, to further emphasize the experimental nature of our study, we have added 'simulated' in the abstract (L38), while specifying 'a range of increasing temperatures' throughout the manuscript. In addition, we have added 'simulated' to all mentions of 'plume' throughout the manuscript to make sure the reader understands that we were not conducting *in situ* experiments. To highlight some of our edits, we have listed the changes (i.e. underlined) at the start of each manuscript section:

- (Abstract L37) 'Here, we investigate the effects of simulated ocean warming and sediment plumes on the cosmopolitan deep-sea jellyfish *Periphylla periphylla* ...'
- (Introduction L135) 'Here, we investigate the effects of simulated ocean warming and mining-induced sediment plumes on the pelagic helmet jellyfish *Periphylla periphylla*.'

- (Results L154) ‘In total, 64 *P. periphylla* were collected from the Lurefjord (n=38) and Sognefjord (n= 26), of which 21 were experimentally exposed to a range of increasing temperatures and 43 to different concentrations of suspended sediment.’
- (Results L361) ‘Response to simulated sediment plumes’.
- (Results L362) ‘Simulated sediment plumes showed a marked effect on *P. periphylla*’s physical condition.’
- (Discussion L497) ‘In this study, we investigated the response of a deep pelagic jellyfish to simulated ocean warming and mining-induced sediment plumes, ...’

Page: 4

L123 regarding ‘global warming’: Same comment as previously in abstract. Please amend.

Adjusted as described above (L135).

L129, regarding ‘increasing temperatures’: a range of temperatures

Adjusted as follows (L142): ‘to a range of increasing temperatures’. Moreover, we have changed the use of ‘increasing temperatures’ throughout the manuscript.

L130: I don't see any behavioural studies...

Since *P. periphylla* produced mucus at different rates in response to our sediment treatments, we characterized this response as ‘behavior’. We do acknowledge that it is a matter of semantics whether or not mucus production in jellyfish can be considered as behavior (especially if relating it to consciousness) and have therefore removed ‘behavior’ from the sentence. The response is now described by ‘physiological data’ (L143). The same was done in the abstract (L39) and discussion (L498, 748).

L131: This summary sentence seems a bit clumsy and does not include the condition metrics. Suggested rewording: Specifically, we assessed metabolic responses to temperature and sediment as well as effects of treatment conditions on *P. periphylla* microbiome. Condition scores were also assigned after exposure to sediment treatments.

We agree that readability can be enhanced by making the sentence more concise and have made the following changes:

(L149) ‘Specifically, we measure their stress response based on physical condition, respiration, ammonium excretion, gene expression and changes in associated microbiota.’

Page: 5

L137, regarding ‘increasing temperatures’: replace with ‘a range of’

Adjusted.

L138, regarding ‘plumes: Delete - they are not being exposed to plumes as this would require in situ experiments. Similar references to plumes should be removed throughout

Adjusted throughout the main manuscript and supplementary methods.

Page: 6

Fig 1C, regarding sample IDs: these numbers should be explained or deleted. I assume they are simply different individuals?

Adjusted by describing numbers (in Fig. 1, L180 and Fig. 2, L353) ‘Columns indicate sample numbers.’

L147, regarding species name: spell in full

Adjusted throughout manuscript and supplementary methods.

L148, regarding ‘fiber optics’: oxygen optodes

Adjusted and in L895 (methods).

L149, regarding summary statistics: Not needed in figure caption

Removed F statistics and degrees of freedom, but kept statistical method and p-value as this is the same syntax used in other Nature Communication papers. The same was done for figure caption 2.

L149, regarding heatmap figure and caption: Explain color coding. The z score is not explained in the results and reference to Fig 1C is missing. Please amend

To align with Nature Communication formatting (e.g. Hurskainen et al. 2021. *Nature Communications*, Ceaser et al. 2022. *Nature Communications*), the explanation of the Z-score was kept in the methods (L982), and was amended in the Figure 1 and 2 captions as follows (L180 and L354):

‘Each row represents one gene, with colors showing z-score-transformed expression values where blue indicates below- and yellow above-average expression..’

Page: 7

L159 regarding ‘in respiration rates’: add ‘among treatments’

Adjusted.

L173 regarding ‘complement activation’: I assume this means the upregulated genes are indicators of activation of innate immunity but this isn't clear, please re-word.

To better clarify, the sentence was adjusted as follows:

(L222) ‘Our gene ontology (GO) analysis further confirmed that the significantly upregulated genes were enriched in processes related to innate immunity (i.e. activation of the complement system by Q7SIC1, a defensive fuclectin; and O75369, a filamen-B), ...’

L182 regarding ‘KEGG’: Define or explain what this is

Thank you for pointing this out, we have made the following edit to better explain the first mention of KEGG:

(L218) ‘By assigning biological function through the Kyoto Encyclopedia of Genes and Genomes (KEGG)³⁸, functional categories of overexpressed genes were found to be associated with processes such as ribosome translation, the immune system (i.e. C-type lectin mediated by Q9NBX4, a reverse transcriptase), and cell growth and death (Supplementary Table 3, T1).’

Page: 8

L204: This was confusing at first - there was no logical reason why the day of the experiment would matter. In the discussion, the text mentions day of COLLECTION was different. If this is accurate, please change here to reflect that.

We agree that this section would benefit from the suggested distinction and have made the following changes, in addition to updating the supplementary table captions to Supplementary Tables 4 and 5:

(L266, Response to temperature) ‘When comparing the microbial community structure of *P. periphylla* across treatments, jellyfish microbiomes were significantly different from the sea water (PERMANOVA $df=1$, $F=23.20$, $p=0.001$, $n=24$) with no effect of the temperature treatments (Supplementary Fig. 4a, Supplementary Table 4). Nevertheless, the sampling date proved to significantly affect microbial composition (PERMANOVA $df=1$, $F=8.28$, $p=0.004$, $n=12$) since not all *P. periphylla* were collected on the same date due to logistic limitations aboard the research vessel. Consequently, microbiomes of individuals exposed to 9.5°C could not be compared to those at 7.5 and 11.5°C, as individuals in the former treatment were collected on a different day.’

(L431, Response to suspended sediment): ‘Microbial community structure of *P. periphylla* in response to sediment plumes appeared to be differently affected among our two sampling days and thereby experimental runs.’

Page: 9

L223: Cross-Out ‘plume’

Changed to ‘simulated plumes’ (i.e. see response to reviewer comment starting with Page 2, L38).

L224 regarding ‘plumes’: concentrations

To ensure clarity of our terminology, we have changed ‘sediment plume’ to ‘suspended sediment’(L343).

L227 regarding summary statistics: Dont need stats in captions

Adjusted as described above to comply with Nature Communications syntax.

L229 regarding heatmap figure: Same comments as for 1C.

Adjusted as described above.

Page: 10

L236 Cross-Out ‘deep sea mining plumes’: replace with 'sediment exposure - or something similar. Not plumes.

Changed to ‘Response to simulated sediment plumes’ (please see response to reviewer comment starting with Page 2, L38).

L237: Cross-Out ‘plumes’

Changed to ‘Simulated sediment plumes’ (as described in response to reviewer comment starting with Page 2, L38).

L238 regarding ‘could not be supported statistically: Needs re-wording. doubling was observed, between 1 and 333 mg treatments, but was not statistically different.

Adjusted as follows:

(L363) ‘..., while the doubling of respiratory rates for individuals in the 0 and 333 mg L⁻¹ treatments could not be supported statistically.’

L241: Cross-Out ‘just’

Adjusted.

L 242: Cross-Out ‘to which’: replace with ‘and’

Adjusted.

L248: Cross-Out ‘from’: replace with ‘than’

Adjusted.

L251: Cross-Out ‘plume’: replace with ‘sediment’

Adjusted.

Page: 11

L285: was affected by experimental (or collection?) day. See earlier observation regarding experimental day vs collection day (L201)

Adjusted as described above:

(L375) ‘Microbial community structure of *P. periphylla* in response to suspended sediment appeared to be differently affected among our two sampling days and thereby experimental runs.’

L291: Cross-Out ‘Interestingly’: Subjective observation

Adjusted.

Page: 12

L300: Cross-Out Date: ‘plumes’

Adjusted.

L305: why the different number of individuals in each treatment? Needs to be explained in the methods. While we aimed to have equal sample numbers, our sample sizes were ultimately determined by the quantity and size of jellyfish caught and the capacity of our experimental tanks to give the animals enough space for normal behavior. Consequently, experimental tanks comprised one large or up to four small jellyfish, to which treatments were assigned randomly. As such, overall sample sizes per treatment were dependent on the availability of caught specimens, while we unfortunately lost three microbiome samples as these did not yield DNA during microbiome extractions. We agree that this should have been made more clear in the methods and have made the following changes in the Methods, including linking Table S1 and our PANGAEA dataset for more detailed data on the size distribution across treatments:

(886) ‘Sample sizes for each experimental run were determined by the quantity and size of jellyfish caught to allow enough space for normal behavior, similar to that observed in individuals *in situ*⁷⁰, in each of the five experimental tanks. As such, kreisel tanks always contained up to four small (0.83–5.33 cm CD) or one large (6.40–10.54 cm CD) *P. periphylla*. Each of the five sediment concentrations were randomly assigned to kreisel tanks, resulting in roughly similar size distributions across treatments (i.e. on average ranging between 2.96 and 4.09 cm in CD, see Supplementary Table 1 or the PANGAEA dataset doi:10.1594/PANGAEA.957367).’

L308: Cross-Out ‘it is worth noting that’

Adjusted.

Page: 13

L312 regarding Fig 3A treatment names: Change to ‘sediment’ no ‘plume’

L315: Cross-Out ‘plume’: change to ‘sediment’

Regarding both comments to Fig. 3, ‘plume’ in the figure legend was kept but in the figure caption changed to ‘simulated plume treatments’ (L485), which should now clarify any ambiguities (i.e. see response to reviewer comment starting with Page 2, L38).

Page: 14

L325 regarding ‘investigate’: investigated

Adjusted.

L325 insert between ‘to’ and ‘ocean’: conditions expected under future

Adjusted as follows (L497): ‘In this study, we investigated the response of a deep pelagic jellyfish to simulated ocean warming and mining induced sediment plumes, ...’

L326 insert between ‘and’ and ‘mining’: during

Left as is, as we here meant the ‘response to’ mining-induced sediment plumes.

L326 regarding ‘behavior’: unclear which of these experiments is behaviour. If the authors mean the health status observations, this seems more physiological than behavioural but if this is what the authors are referring to, they should explain/clarify

Adjusted as described above (see response to reviewer comment starting with Page 4, L130).

L828: Cross-Out 'worrisome' and 'healthy'

Both adjusted as suggested.

L330 regarding 'provisioning': provisioning of food?

Correct. To better clarify, we changed the sentence as follows (L502): '... and the provisioning of food for commercially important fish stocks.'

L331 regarding 'stocks': such as?

Since this the introductory summary statement of our Discussion, we have chosen not to elaborate on the fish stocks here, and instead made the following amendment in the introduction:

(L113) '... as deep pelagic fauna play a vital role in the provisioning of commercially important fish stocks such as tuna, ...'

L335: Cross-Out 'appeared to be less pronounced statistically speaking': was not statistically significant

Adjusted as suggested.

L338: This statement needs more explanation. Organisms usually use anaerobic respiration when they have to - ie under low o2 conditions. Were your experiments conducted under low o2? Just increasing temp does not imply conditions are low enough to stimulate anaerobic resp. Maybe this dual simultaneous respiration is a characteristic of jellyfish physiology, but if this is the case, a more in depth explanation of the observed results is needed in the context of the different respiratory pathways.

Thank you for pointing this out, a similar point was raised by reviewer 1. Please find our copied response below:

Thank you for your suggestion and thoughts on the matter. A similar point was raised by reviewer 2 and we agree that the manuscript will benefit by better highlighting the natural variability of oxygen consumption data (as mentioned in Thuesen and Childress, 1994. *Biological Bulletin*, and Youngbluth and Båmstedt, 2001. *Hydrobiologica*) and reflecting this in our discussion of aerobic respiration. To this end, we have made the following changes to the paragraph:

(L507) 'This moderate response in oxygen consumption can potentially be explained by *P. periphylla*'s ability to perform both aerobic and anaerobic respiration simultaneously³⁹, with downregulated genes here suggesting a shift away from aerobic metabolism. Although we did not find any significantly DE genes directly related to anaerobic metabolism (e.g. involving glycolysis), anaerobic respiration may have been continually active in *P. periphylla* across treatments. Consistent levels of anaerobic enzymes were also found in *P. periphylla* populations living at low versus higher oxygen concentrations, while those in low oxygen environments carried significantly lower levels of aerobic enzymes³⁹. Moreover, *P. periphylla*'s ability to respire anaerobically may contribute to the overall low respiration rates, which were within the range of previously reported values for *P. periphylla* and similar to those of other deep pelagic medusa capable of anaerobic respiration (Supplementary Table S7)^{35,40}. Nevertheless, further research is required to determine whether oxygen concentrations and seawater temperature have similar effects on aerobic versus anaerobic metabolism in *P. periphylla*. Alternatively, the response in oxygen consumption may still become significant when sampling sizes are increased, considering the natural variability in these animals and associated data^{35,40}. However, based on our current results and given that *P. periphylla* can vertically migrate to significantly warmer waters at night and has a wide geographic distribution^{34,35}, we expect a relatively wide temperature tolerance in this species. Moreover, when considering current climate warming projections (i.e. approximately 1°C over the next 84 years, with 4°C warming only

under extreme climate predictions)⁴¹, global warming scenarios for the deep ocean do not appear to pose an immediate respiratory threat for *P. periphylla*.’

L339: Cross-Out ‘Interestingly’

Adjusted.

L341: this seems to support your earlier statement but still doesn't explain how the different pathways operate under different conditions - or why they would both operate under oxic conditions

Adjusted, please see response under reviewer comment above, starting with Page 14, L338.

L343: this makes sense - but does this apply to your experimental conditions?

Adjusted, please see response under reviewer comment above, starting with Page 14, L338.

L348: I don't follow the logic relative to your experimental conditions.

Adjusted, please see response under reviewer comment above, starting with L338.

L352: The fjords you made collections for your experiments had different temperature profiles so although they were similar temps at 300m, if collections were performed from 0-800 m (as described in the methods) it is possible some of the jellyfish came from waters as warm - if not higher - than your upper experimental level. The results are not presented separately for the different fjords - this potential artefact should be discussed somewhere

Since the jellyfish were collected over a depth interval, they could indeed have been collected from different depths. To account for this, all jellyfish were acclimatized at the same temperature for 7 to 14 hours before being exposed to any treatments. We recognize that their migratory behavior is likely to influence their thermal tolerance, which is why we have raised this point here in this and the following paragraph of the Discussion (in addition to showing the temperature profiles across fjords in the Supplementary Materials).

Page: 15

L357: Cross-Out ‘Interestingly’

Adjusted.

L357-358: I suggest moving this statement to line 364, in front of 'moreover'

Kept as is since this is the topic sentence of the paragraph

L358: Cross-Out ‘appeared to be’ and insert ‘was’

Kept as is, since we here argue that processes related to chemokines, cell adhesion and cell migration may also be linked to innate immunity. The use of ‘was’ is too strong of a statement.

L379 regarding ‘ In any case’: In summary?

Meant to refer to the fact that we currently do not know whether the innate immune response was induced by tissue integrity, pathogens, or perhaps a form of frontloading. To clarify, adjusted to ‘In any event’ (L629) with the following adjustment to make the sentences on tissue integrity, pathogens and frontloading more concise :

(L624) ‘While warming temperatures may compromise *P. periphylla*’s tissue integrity or increase susceptibility to pathogens, it is also possible that overexpression of immune-related compounds is a form of ‘frontloading’ to prepare for regularly encountered stress⁵¹.’

L386: Insert between ‘may’ and ‘specifically’: Be

Adjusted.

Page: 16

L391: What is the implication of this? respiration does not increase, but Ammonium excretions does...so they can adjust respiration rates but not other metabolic responses

Since ammonium excretion is a byproduct of metabolism, both respiration and ammonium are invariably linked. Nevertheless, both metabolic processes can scale differently with increasing temperature depending on the fraction of carbohydrate, protein or lipid storage mobilized, as shown by Shimauchi & Uye (2007. *Journal of Oceanography*) in the coastal jellyfish *Aurelia aurita*. Since we currently do not fully understand the mechanisms of anaerobic and aerobic respiration in *P. periphylla* and can only speculate on the matter, we have chosen to not include this in the manuscript. (NB the entire paragraph on ammonium excretion was moved up to L597 to follow the paragraph that first mentions ammonium excretion.

L397: Amend previous statements about experimental date - this is the relevant information that can explain the microbiome results

Adjusted as suggested.

L441 regarding 'shear': what does this mean in this context?

To better clarify, 'shear' was adjusted to 'mechanical stress' (L659).

L413: Cross-Out 'instance' and insert 'example'

Adjusted.

Page: 17

L430: Cross-Out 'notion of'

Since this suggestion is a matter of writing style and does not change the meaning of the sentence, we chose to keep this part of the sentence as is.

L431: Cross-Out 'On the other hand' and insert 'Alternatively,'

'On the other hand' is meant to signal contrast to the previous sentence mentioning similarities in the transcriptomic response of *P. periphylla* in response to temperature. 'Alternatively' would signal an alternative explanation, which is not what is meant here. As such, the sentence is kept as is.

L442: Spell genus

Adjusted.

L443: spell genus

Adjusted.

L449: couldnt it also mean Pp has the capacity to adapt to or repair damage - which has a potential energetic cost, but that would be an indirect effect

Thank you for your suggestion, we completely agree and had discussed this in the last paragraph of the Discussion, starting on L783. Since the current paragraph is meant to discuss direct effects of the transcriptomic response, we have left this section as is.

L453: Cross-Out 'Interestingly': Subjective

Changed to 'surprisingly' to signal that the result was unexpected.

Page: 18

L465: Were these in situ or ex situ experiments. If the latter, they were not exposed to plumes. Please amend if necessary

Changed to 'simulated sediment plumes ex situ'.

L469 regarding 'behavioral': same question as previously

Adjusted.

L477 regarding ‘could not be annotated through homologous function’: Not sure what this means - explain

We meant annotation based on sequence homology, where gene function may be assigned by identifying evolutionarily related genes. To better clarify, we have replaced homologous function with ‘sequence homology’ (L756).

L484: Cross-Out ‘one of our most worrying findings is that’

We recognize that the current formulation is subjective and have changed the sentence as follows: (L774) ‘... it should be stressed that *P. periphylla*’s metabolic response to sediment plumes exceeds the most extreme warming treatment.’

L486: Cross-Out ‘plume’

Adjusted.

L490 regarding ‘offset energy demands’: I don't think this is quite the right wording. My interpretation is that prolonged sediment exposure can shift energy budgets and affect long term fitness because energy intake could not compensate

The current sentence is part of our argument that sediment exposure could indeed affect energetic budgets and fitness (L783). To better clarify, we have moved the current sentence down with the following amendments:

(L780) ‘Although the exact effects and metabolic costs will depend on suspended sediment concentrations, our results suggest that prolonged sediment exposure has the potential to offset energy demands that have taken millions of years to evolve¹³, even at low concentrations. Specifically, if mining plumes are present in the water column for extended periods of time, elevated metabolism has either to be met with increased food intake, or sustained mucus production will lead to energetic depletion and thereby a reduction of fitness and possibly death.’

Page: 19

L495 regarding ‘physical exhaustion’: energetic depletion?

Adjusted.

L512: This covers the surface temperatures which are different from those at 300m. Also, why so deep when the distribution graph shows distribution at night is <300m?

Since we did not employ nets with opening and closing systems that could potentially damage jellyfish, we automatically sampled surface waters. To account for this, all jellyfish were acclimatized at the same temperature before being exposed to any treatments (i.e. please also see response under reviewer comment above, starting with L352). Some nets were deployed deeper than 300 m as we did not always collect jellyfish with each haul and wanted to ensure to collect jellyfish that may potentially also be in deeper waters. Exact collection depths are featured in the PANGAEA dataset, to which we have now made the following reference:

(L801) ‘For experiments, individuals were caught between 0 and 800 m depth after sunset (i.e. for precise collection depths see the PANGAEA dataset doi:10.1594/PANGAEA.957367), ...’

L515 regarding ‘300 m depth’: why this depth?

Sentence amended as follows:

(805) ‘Water for the experimental tanks was collected below the thermocline with Niskin bottles attached to a CTD rosette at 300 m depth, to avoid catching copepods that were present at shallower depths.’

L525: reference fig S8 here - but this temp is the in average in situ temp at 300m between the two fjords, not necessarily the temp at which the specimens were collected - include this information.

To better clarify, the sentence was adjusted as follows:

(833) ‘Individuals that were undamaged after net sampling and therefore suitable for experiments were acclimatized for 7 to 14 hours in seawater at their average *in situ* temperature below the thermocline (7.5°C; Supplementary Fig. S8).’

L526 regarding ‘crates’: crates generally have holes in them - do you mean tanks?

Adjusted.

Page: 20

L527: Cross-Out ‘plume’

Changed to ‘simulated plume’ (i.e. see response to reviewer comment starting with Page 2, L38).

L530: how were these temperatures selected?

We recognize that the selected temperatures require more explanation and added the following sentences: (L842) ‘Since the *P. periphylla* from this study were obtained from coastal fjords and undergo shallow diel vertical migrations during winter, they are naturally exposed to temperature changes of 2–4°C. Water temperatures were therefore not only chosen to mimic ocean warming scenarios (i.e. ~1°C over the next 84 years, with 4°C warming only under extreme climate predictions)⁴¹, but also to estimate temperature-dependence of *P. periphylla* metabolic activity across equally increasing temperature treatments.’

L531 regarding ‘medusae’: this is the only reference to medusae vs jellyfish - define for readers that are unfamiliar with the taxa - or just use jellyfish as before

To indicate that both terms are near-synonymous, we had referenced the terms ‘jellyfish’ and ‘medusa’ twice in the final paragraph of our introduction (L136 + L141). Consequently, we have therefore decided to leave the section as is.

L532: more information is needed on experimental design. Since you are using parametric statistics, the ideal design would have equal replicates per treatment but this doesn't seem to be the case from the graphs and descriptions of results. It also seems that multiple runs were performed rather than each experiment completed at the same time -but again, this is unclear.

Equal sample sizes is actually not one of the assumptions for a one-way ANOVA. However, we agree that we should have explained our tests for normality and have added the following to the ‘Statistical analysis and Reproducibility’ section:

(L1017)‘For each analysis, the normality of residuals and homogeneity of variances were assessed using the Shapiro-Wilk and Levene’s tests, respectively.’

To address our choice of sample sizes and splitting treatments across days, we have added the following sentence to the current paragraph:

(L847) ‘Sample sizes for each temperature treatment were determined by the quantity and size of jellyfish caught, in addition to the availability of our respiration chambers, which occasionally meant splitting our treatments between days (i.e. see Supplementary Table 1 or the PANGAEA dataset doi:10.1594/PANGAEA.957367).’

L535: Cross-Out ‘discontinuously’

Adjusted.

L538: Cross-Out ‘always’

Adjusted.

L539 regarding ‘continuously’: data are always collected at a specified time interval even if very frequently, its not continuous.

Adjusted.

L549 regarding ‘range of distances from a mining site’: needs reference

Sentence split into two to better clarify choice of sediment concentrations:

(L874) ‘To determine the effect of sediment plumes on *P. periphylla*, jellyfish were exposed to five sediment concentrations (i.e. 0, 16.7, 33.3, 166.7, 333.3 mg·L⁻¹) at their average *in situ* temperature (7.5°C) for 24 hours. Sediment concentrations were chosen to represent a range distances from a mining site, as plume dynamics will be highly dependent on the quantity of released sediment, sediment type and hydrographic features^{20,21}.’

L552: insert ‘/’ within ‘RV’

Adjusted throughout the manuscript.

L556 regarding ‘four small or one large’: Explain experimental design - how many of each size were included in the experiment and discuss how any unequal size/number distribution may affect your results.

Adjusted as follows (please see response to reviewer comment starting with L305):

(886) ‘Sample sizes for each experimental run were determined by the quantity and size of jellyfish caught to allow enough space for normal behavior, similar to that observed in individuals *in situ*⁷⁰, in each of the five experimental tanks. As such, kreisel tanks always contained up to four small (0.83–5.33 cm CD) or one large (6.40–10.54 cm CD) *P. periphylla*. Each of the five sediment concentrations were randomly assigned to kreisel tanks, resulting in roughly similar size distributions across treatments (i.e. on average ranging between 2.96 and 4.09 cm in CD, see Supplementary Table 1 or the PANGAEA dataset doi:10.1594/PANGAEA.957367).’

L558: this means your exposures treatments will overlap - need to include this artefact in the discussion - ie your treatment exposures only lasted up to 6 hours not 24

Thank you for your suggestion, a similar point was raised by reviewer 1. Please see below our copied response.

Although we suspended a given amount of sediment in the tanks, the concentration did decrease over time due to sedimentation. Therefore, the animals were not exposed continuously to the nominal concentration, but a gradually and substantially lower one. Nominal concentrations in the higher treatments are comparable to those near the initial sediment discharge (Muñoz-Royo et al. 2021. *Communications Earth & Environment*), but the exposure is more in the range of the actual plume. To address the effect of the sediment dilution, we have added the following to the Discussion and Methods:

(L677 Discussion): ‘In addition, we should emphasize that the exact concentrations of sediment plumes generated by a mining operation will be highly dependent on the initial amount of sediment discharged, the sediment type and hydrographic features²⁰⁻²². Choosing the right concentrations of sediment exposure is therefore open to further improvement. In spite of this, it is worth noting that our sediment treatments not only mimic a range of distances from an initial sediment release site, the settling of sediment in our experimental tanks also represents *in situ* sedimentation and dilution of a plume and places the overall sediment exposure in the ranges of experimentally measured plume concentrations²⁰.’

(L793 Methods): ‘As such, the sediment treatments also represent a range of initial discharge concentrations, with the dilution and sedimentation of mining plumes mimicked by the settling of

sediment in the experimental tanks (i.e. decreasing 50% of total after 4-6 hours of incubation, Fig. S10, measured with a Turb® 340 IR meter).’

Page: 21

L580: why not just put them straight in -80?

Since the *P. periphylla* were relatively big, they were first snap frozen at -20°C to limit warming of the -80°C freezer. The sentence was adjusted as follows:

(L926) ‘To limit the effect of handling on jellyfish metabolism, whole jellyfish were first snap frozen at -20°C to minimize warming of the -80°C freezer and then transferred to -80°C within 5 minutes of being taken from their experimental containers.’

L584: Cross-Out ‘back ashore and insert ‘on shore’

Adjusted.

L585: why?

Adjusted as follows:

(L932) ‘Only samples from the HE570 cruise were used for microbiome and transcriptome analysis due to project timing constraints.’

Page: 22

L617:why?

Since we could only generate data for a single replicate in this treatment, we were not able to reliably assess the effect of treatment versus biological variability in this sample, especially in comparison to the other treatments that had >3 replicates each. To better explain, the sentence was adjusted as follows:

(L962) ‘Since *P. periphylla* in the 33 mg L⁻¹ treatment consisted of a single replicate, and we could not reliably assess the effect of treatment versus biological variability, this treatment was not included in downstream analysis.’

Page: 23

L629: Cross-Out ‘in’

Adjusted.

SUPPLEMENTARY MATERIALS

Fig S1 caption: spell out genus in figure captions. Cross-Out ‘plume’

Adjusted for all figures.

Fig S2 caption: Cross-Out ‘plume’

Adjusted throughout supplementary methods.

Fig S5 caption, regarding ‘before admission’: replace with ‘introduction of’

Adjusted.

Fig S6 caption: Cross-Out ‘plumes’

Adjusted throughout supplementary methods.

Fig S7 caption: text states sampling was done from 0-800m - why so deep if the maximum depth is 350 m?

See copied response to reviewer comment starting with L512:

Some nets were deployed deeper than 300 m as we did not always collect jellyfish with each haul and wanted to ensure to collect jellyfish that may potentially also be in deeper waters. However, most of the individuals were collected from <400 m depth and the exact collection depths are featured in the PANGAEA dataset. To better specify this, we have now made the following reference in the manuscript: (L720) ‘For experiments, individuals were caught between 0 and 800 m depth after sunset (i.e. for precise collection depths see the PANGAEA dataset doi:10.1594/PANGAEA.957367), ...’

Table S1 regarding ‘plume’: change to sediment throughout

Adjusted throughout the table.

Table S7 caption: spell out genus

Adjusted for all tables.

Reviewer #3 (Remarks to the Author):

I was asked to review the omics approaches detailed in the manuscript. Overall, the methodology employed is commonly used state of the art for the RNAseq and Microbiome pipeline. However, magnitude of molecular responses observed makes me question the interpretation of the data; the authors state 'worrisome implications' when in fact gene expression and microbiome response are marginal (physiological data set aside).

Thank you for sharing your insights and suggestions, we value your feedback and agree on several points, which are actually in agreement with the current interpretation of our data. First, the microbiome response is indeed not significant, and *P. periphylla*'s microbiome is clearly not affected by short term exposure to warming temperature and suspended sediment. This is in line with what is known for other cnidarian jellyfish, that possess bottom-up control over their microbiome through the expression of mucus and anti-microbial proteins (i.e. reviewed by Tinta et al. 2019. *Marine Drugs*, and discussed in the manuscript). Second, we recognize that the response to warming temperatures appears marginal, mainly supported by physiological data, which in turn helped us interpret the gene expression results (e.g. upregulation of innate immunity is perhaps simply a form of frontloading to prepare for temperature changes). These results seem to fit the fact that *P. periphylla* can vertically migrate in the water column (where they are naturally exposed to temperature changes of 2–4°C), which gives them a relatively wide temperature tolerance. The results for the simulated sediment plumes, on the other hand, give much more reason for concern, especially in the light of our physiological data (i.e. excess mucus production) which should be taken into account when interpreting molecular data. We do agree that our phrasing of 'worrisome' is subjective (a similar point was raised by reviewer 2), and we have removed this and similarly subjective adjectives throughout the manuscript.

Sequencing data is not available on NCBI under the provided BioProject IDs; these data must be made available prior to any publication. Also, it is nowadays fairly common to provide the bioinformatic workflow, e.g. via GitHub as well as the assembled transcriptome. Both are not made available [This is even more curious, since physiological and microbiome data are provided]. Thus, the data produced in the manuscript is of limited use to other researchers.

We hope that the reviewer will understand our concern of making the sequences publicly available prior to publication. As such, we had generated the following reviewer-links that give selective access to all sequencing data:

Transcriptomes:

<https://dataview.ncbi.nlm.nih.gov/object/PRJNA971902?reviewer=bbq7mm5rvqkoopc9s1qushhma3>

Microbiomes:

<https://dataview.ncbi.nlm.nih.gov/object/PRJNA971258?reviewer=u2gu5a86r3vm8fecrk9s7rcn2>

All sequencing data will be made publicly available together with the publication of the manuscript. If the reviewer would like to see the complete sequencing data prior to publication, we are happy to share all sequences privately via dropbox.

Moreover, we have now also uploaded the assembled transcriptome to the Figshare database under the following reviewer-link: <https://figshare.com/s/8251a92387411dbb7ebc>, and submitted our normalized expression counts as part of our differential gene expression analysis to the PANGAEA database (i.e. also

including log₂ fold change values of significantly DE transcripts and annotations of expressed transcripts; <https://doi.pangaea.de/10.1594/PANGAEA.962217>).

Regarding the bioinformatic workflow, the microbiome pipeline published by Bush et al. (2022. *Nature Communications*) on Github was followed (as referenced on L905 ‘The bioinformatic pipeline published by Busch *et al.*⁸⁴ was followed for analysis of generated sequences ...’). For the transcriptomics, we used publicly available software with well supported Github pages, including instructions, which is why we haven’t released a separate Github page as we would merely be repeating their terminal commands. That said, we are naturally more than happy to share any scripts if requests are sent to the corresponding author.

I was pretty surprised to see the very high number of assembled transcripts without any discussion. At large, it is rather hard to follow how things were done with how many replicates and what cutoffs. A flow-chart as a supplemental figure and a supplemental table detailing the numbers would be a huge help.

It is actually quite common for *de novo* assemblers to produce many more transcripts than expected based on the number of genes in a genome (i.e. as reviewed by Raghaven et al. 2022. *Briefings in Bioinformatics*). This is generally caused by the inclusion of gene isoforms (from alternative splicing), pre-mRNA, ncRNA or fragmented transcripts during RNA extractions. To be sure of our assembly method, we actually also ran a *de novo* assembly with RNAspades, that returned a similar number of transcripts and showed similar BUSCO-completeness. Since Trinity is the most prominent assembler (as reviewed by Raghaven et al. 2022), and comparison of assembly software was not the focus of our manuscript, we chose to only include the Trinity assembly.

To obtain the number of transcripts representing putative protein coding genes, we filtered out all non-expressed contigs, which gives a conservative estimate of gene number (Raghaven et al. 2022). Moreover, our finding of ~21.000 putative genes in *P. periphylla* is within the range of other jellies (e.g. *Aurelia aurita* ~30.000; Gold et al. 2019. *Nature Ecology & Evolution*).

We recognize that this should have been explained better in the manuscript and we have therefore added the following sentence to the Results, which also addresses the use of ‘genes’ versus ‘transcripts’: (L161) ‘Within the transcriptome (also containing gene isoforms and non-coding RNA), up to 21,429 transcripts were actively expressed, which we hereafter refer to as putative genes.’

I have to say that I find the differential gene response marginal for the temperature treatment (167 up and 54 down-regulated out of 20,697 putative genes < 1%; the change from 839,298 transcripts to 20,697 is also not detailed/documented/discussed). Is this tiny response expected? what is the average fold change? Why is it stated 'genes' here when transcripts were assembled? I also find the up regulation of metabolism under increased temperature less than surprising .. this relates pretty much to any organism.

As detailed above, we agree with the marginal response in the temperature treatment, which can be explained by *P. periphylla*’s ability to migrate vertically in the water column. The average log₂ fold change for significantly upregulated genes in the temperature experiments is 2.4 (max =6,7) and -3.4 (min = -7.9) for downregulated genes, which are accepted values to signal significantly over- and under-expressed genes. While the regulation of metabolism under increased temperature may be less surprising, we here show this response (including innate immunity) for the first time in a deep pelagic jellyfish,

Nature Communications, Stenvers & Hauss et al.

making this study a starting point to unravel their underlying cellular stress response.

REVIEWERS' COMMENTS

Reviewer #1 (Remarks to the Author):

The authors have done a very nice job of addressing my initial comments. In this regard I have only one minor additional comment (see below). Their paper is a much needed contribution to the literature and their conclusions, now tempered somewhat, are fully supported by their data. For instance, the authors make many appropriate caveats about whether their results for this jellyfish may be representative for other taxa (e.g. line 726 - "If *P. periphylla*'s response is representative of other gelatinous organisms, a diverse and abundant group of animals in the deep ocean³, the commercial mining of the deep-seafloor may impact biodiversity and the healthy functioning of the deep ocean, including its many services.")

line 818-820 - Models suggest settling of sediment but exactly how long a jelly might be exposed to a given concentration is uncertain as we have minimal in situ data. Thus the wording should be changed to something like

"...with the dilution and sedimentation of mining plumes similar to the settling of sediment in the experimental tanks (unclear durations of exposure in situ, decreasing 50% of total after 4-6 hours of incubation in our experiments....."

Reviewer #2 (Remarks to the Author):

The revised paper is much improved and I am fine with moving on to publication. I look forward to seeing this in print.

Reviewer #3 (Remarks to the Author):

The comments are addressed in a satisfactory manner, although I do not share the worries regarding making data available prior to publication.

I don't quite agree on the response regarding my concerns of the very high number of assembled transcripts. Of course I am aware of everything the authors state here, but in all

earnest, the response is rather generic and does not detail how the authors actually addressed to come from >800,000 to ~ 20,000 loci ..

The authors state "To obtain the number of transcripts representing putative protein coding genes, we filtered out all non-expressed contigs, which gives a conservative estimate of gene number (Raghaven et al. 2022). Moreover, our finding of ~21,000 putative genes in *P. periphylla* is within the range of other jellies (e.g. *Aurelia aurita* ~30,000; Gold et al. 2019. *Nature Ecology & Evolution*)."

so the above procedure removed 780,000 transcripts? how many transcripts do you retain to cover the 21,000 orthologs?

There is also no explanation why trimmed and non-trimmed assemblies feature largely different GC contents. This to me would suggest that you assembled transcripts/genes from other than the target organism. All of this is fine and understood, but it would be nice to have this stated and transparent. For instance, you could run a BLAST and show the taxonomic hit distribution that your assembled transcripts hit.

All I am saying is I am doing a lot of RNASeq assemblies and the numbers strike me as very high, and we typically try to get to the base of this. The responses provided here don't really help me understand how things were filtered or derived. Of course, 20,000 putative genes is great, but it's still unclear how you came to that number coming from 800,000 transcripts.

Response to reviewer comments 2

We highly appreciate the reviewers for their time and suggestions to further improve our manuscript. In the following, we provide point-by-point responses (in blue) detailing how we addressed the feedback by the reviewers. Line numbers refer to the cleaned and final version of the manuscript.

The most important revision includes clarification of the methods on how the final transcriptome was obtained.

Reviewer #1 (Remarks to the Author):

The authors have done a very nice job of addressing my initial comments. In this regard I have only one minor additional comment (see below). Their paper is a much needed contribution to the literature and their conclusions, now tempered somewhat, are fully supported by their data. For instance, the authors make many appropriate caveats about whether their results for this jellyfish may be representative for other taxa (e.g. line 726 - "If *P. periphylla*'s response is representative of other gelatinous organisms, a diverse and abundant group of animals in the deep ocean³, the commercial mining of the deep-seafloor may impact biodiversity and the healthy functioning of the deep ocean, including its many services.")

line 818-820 - Models suggest settling of sediment but exactly how long a jelly might be exposed to a given concentration is uncertain as we have minimal in situ data. Thus the wording should be changed to something like
"...with the dilution and sedimentation of mining plumes similar to the settling of sediment in the experimental tanks (unclear durations of exposure in situ, decreasing 50% of total after 4-6 hours of incubation in our experiments....."

Thank you, we are very grateful for your positive and helpful feedback. We have made the following changes to better highlight uncertainty for in situ exposure times:

(L643 Methods) "... with the dilution and sedimentation of mining plumes mimicked by the settling of sediment in the experimental tanks (i.e. durations of *in situ* exposure unknown, with concentrations here decreasing 50% after 4-6 hours of incubation, Supplementary Fig. 10, measured with a Turb® 340 IR meter)."

Reviewer #2 (Remarks to the Author):

The revised paper is much improved and I am fine with moving on to publication. I look forward to seeing this in print.

Thank you for your positive feedback, we greatly appreciate it.

Reviewer #3 (Remarks to the Author):

The comments are addressed in a satisfactory manner, although I do not share the worries regarding making data available prior to publication.

I don't quite agree on the response regarding my concerns of the very high number of assembled transcripts. Of course I am aware of everything the authors state here, but in all

earnest, the response is rather generic and does not detail how the authors actually addressed to come from >800,000 to ~20,000 loci ..

The authors state "To obtain the number of transcripts representing putative protein coding genes, we filtered out all non-expressed contigs, which gives a conservative estimate of gene number (Raghaven et al. 2022). Moreover, our finding of ~21,000 putative genes in *P. periphylla* is within the range of other jellies (e.g. *Aurelia aurita* ~30,000; Gold et al. 2019. *Nature Ecology & Evolution*)." So the above procedure removed 780,000 transcripts? how many transcripts do you retain to cover the 21,000 orthologs?

There is also no explanation why trimmed and non-trimmed assemblies feature largely different GC contents. This to me would suggest that you assembled transcripts/genes from other than the target organism. All of this is fine and understood, but it would be nice to have this stated and transparent. For instance, you could run a BLAST and show the taxonomic hit distribution that your assembled transcripts hit.

All I am saying is I am doing a lot of RNASeq assemblies and the numbers strike me as very high, and we typically try to get to the base of this. The responses provided here don't really help me understand how things were filtered or derived. Of course, 20,000 putative genes is great, but it's still unclear how you came to that number coming from 800,000 transcripts.

Thank you for your helpful feedback and clarification of the initial concerns. We have released our sequencing data on NCBI, which is now publicly available.

Regarding the number of assembled transcripts, the ~800,000 transcripts were trimmed to ~293,000 by removing non-expressed contigs (i.e. omitting those transcripts which had normalized expression values of zero across all samples). Following this, the final transcriptome count of ~20,000 was obtained by removing additional low-count transcripts, i.e. only keeping those with ten or more copies in at least two or more samples (which is a common pre-filtering step for tissue RNA seq and differential gene expression analysis, e.g. Srinivasan *et al.* 2016. *Nature Communications*). Although the removal of all non-expressed transcripts is a stringent trimming process to eliminate non-coding RNA, pre-mRNA or fragments, it is the most effective way to only retain biologically relevant data (i.e. differentially expressed genes). Above all, it does not affect the interpretation of our data. To better clarify this, we have made the following amendments:

(L725 Methods) "Transcripts were further trimmed by removing those with zero normalized counts across all samples, reducing the de novo transcriptome to 293,318 transcripts. To further eliminate noise within the matrices and obtain the final transcriptome count, normalized counts were filtered to only include transcripts with a count of at least ten in two or more samples."

Regarding the GC content, the difference between the non-trimmed and trimmed transcriptomes can be explained by both the Trinotate-filter (discarding transcripts without an open-reading frame) and the removal of contaminant sequences. Since the trimming method described above does not remove contamination, contaminant sequences were removed separately before and after the initial de novo assembly. Since the second contaminant removal was based on Blast matches of the annotated transcriptome, the method is similar to running Blast for a taxonomic hit distribution. Nevertheless, we realize that the title for Supplementary Table 2 was misleading as it did not include the information regarding contaminant removal, and have therefore made the following changes:

(Supplementary Materials, Table S2 title) “*Periphylla periphylla de novo* transcriptome statistics for the Trinity assembly, in addition to the trimmed Trinity transcriptome, filtered by the presence of open-reading frames identified by Transdecoder¹² and removal of contaminant sequences.”

In addition to highlighting the GC difference in the Methods:

(Methods L716) ‘Removal of contamination was further confirmed by the overall sequence GC content, which was slightly higher after trimming (Supplementary Table 2).’